# Unlocking Multimodal Document Intelligence: From Current Triumphs to Future Frontiers of Visual Document Retrieval in the Era of Large Language Model

## Abstract

With the rapid proliferation of multimodal information, Visual Document Retrieval (VDR) has emerged as a critical frontier in bridging the gap between unstructured visually rich data and precise information acquisition. Unlike traditional natural image retrieval, visual documents exhibit unique characteristics defined by dense textual content, intricate layouts, and fine-grained semantic dependencies. This paper presents the **first comprehensive survey of the VDR landscape, specifically through the lens of the Multimodal Large Language Model (MLLM) era**. We begin by examining the benchmark landscape, and subsequently dive into the methodological evolution, categorizing approaches into three primary aspects: multimodal *embedding models*, multimodal *reranker models*, and the integration of *Retrieval-Augmented Generation* (RAG) and *Agentic systems* for complex document intelligence. Finally, we identify persistent challenges and outline promising future directions, aiming to provide a clear roadmap for future multimodal document intelligence.

## 1 Introduction

Multimodal retrieval, the task of retrieving relevant multimodal information from a large-scale collection using queries that span multiple modalities like text and vision, has become a cornerstone of modern information retrieval (Mei et al., 2025; Zheng et al., 2025a). Historically, research in this domain has predominantly focused on natural image retrieval, targeting datasets of photographs and web images where the primary goal is to match objects, scenes, or holistic visual concepts (Wu et al., 2024a; Arslan et al., 2024). However, both academia and industry begin to turn their attention to a distinct yet ubiquitous data type: **visual documents**[1]. These

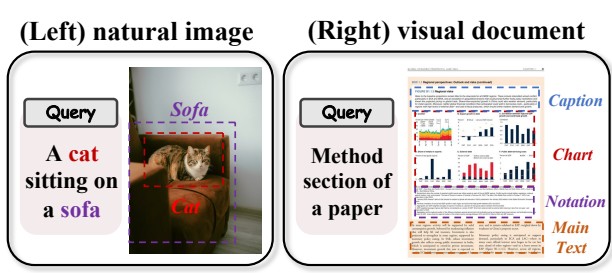

**(Left) natural image**     **(Right) visual document**

**Figure 1:** Comparison of retrieval of natural image (*left*) and visual document (*right*), the focus of this survey.

documents, ranging from scanned PDFs and business reports to invoices and academic papers, are characterized by a dense interplay of textual content, complex layouts, and graphical elements (Tang et al., 2023; Li et al., 2024d). This shift necessitates a exploration of retrieval techniques tailored to the unique structure and information density of visual documents, pushing the boundaries of what multimodal systems can achieve.

The pivot towards **Visual Document Retrieval** (VDR) is driven by three fundamental differences that distinguish visual documents from natural images, as illustrated in Figure 1. ❶ **Information modality and density**: unlike natural images which convey semantic meaning through holistic scenes, visual documents are hybrid entities where meaning is co-determined by rich textual information and a structured spatial

---

[1]They are also commonly referred to as *visually rich documents*, *document images*, *etc.* We use "visual documents" as a unifying term. See the illustrative examples from representative VDR datasets in Figure 2.

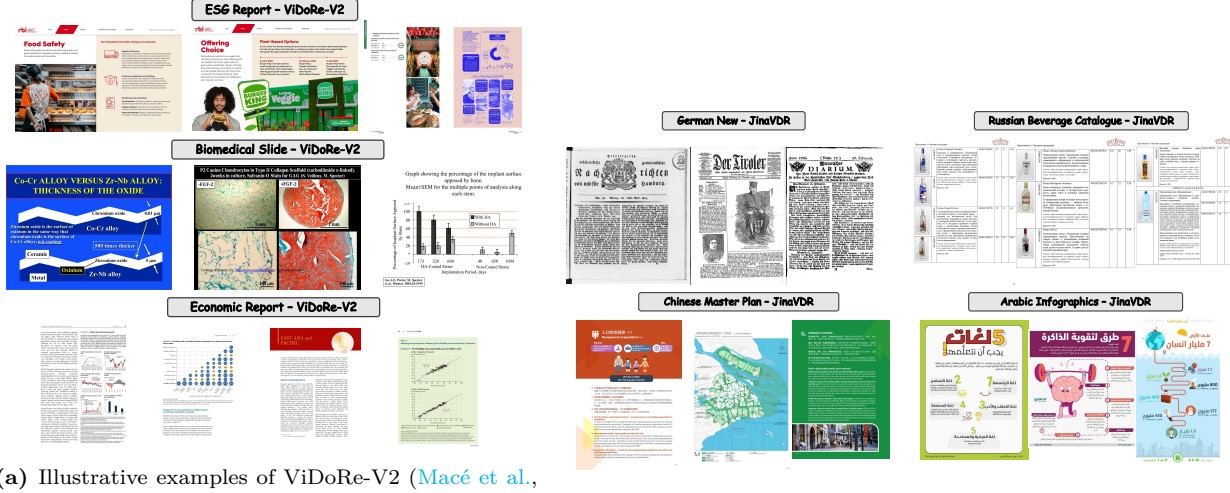

**(a)** Illustrative examples of ViDoRe-V2 (Macé et al., 2025).

**(b)** Illustrative examples of JinaVDR (Günther et al., 2025).

**Figure 2:** Illustrative examples from two representative datasets, ViDoRe-V2 (a) and JinaVDR (b).

layout. The information is dense, hierarchical, and multi-modal by nature. ❷ **Semantic granularity**: retrieval in natural images often targets high-level concepts (*e.g.,* "a cat sitting on a sofa"), whereas VDR demands a much finer-grained understanding. Users may query for specific facts embedded within a table, a particular sentence in a paragraph, or information contingent on its document-level position (*e.g.,* "the methodology section of a paper"). ❸ **User intent and task complexity**: VDR is geared towards precise information-seeking, question answering, and evidence-based reasoning, rather than conceptual or aesthetic matching.

Furthermore, as the general capabilities of Multimodal Large Language Models (MLLMs) advance (Song et al., 2025; Yan et al., 2025b;a), the VDR field is increasingly focusing on their integration. This includes the development of MLLM-based embedding and reranker models to enhance semantic matching (Tao et al., 2024; Zhang et al., 2024b; Wang et al., 2025d). Beyond that, there is active exploration into leveraging these models within more sophisticated frameworks like Retrieval-Augmented Generation (RAG) pipelines (Gao et al., 2023; Cheng et al., 2025; Gan et al., 2025a) and Agentic systems (Singh et al., 2025) to tackle complex document-based settings.

**Scope.** While several surveys have touched upon related areas (as summarized in Table 1), a comprehensive analysis of VDR in the LLM era has been conspicuously absent. Previous reviews have largely concentrated on either traditional information retrieval (Alaei et al., 2016) and general deep learning for document understanding (Subramani et al., 2020; Sassioui et al., 2023; Ding et al., 2024), or retrieval for natural images (Zhou et al., 2017; Hameed et al., 2021). More recent surveys that acknowledge the rise of MLLMs have continued this trend, focusing on general document understanding (Huang et al., 2024; Rombach & Fettke, 2025; Gao et al., 2025; Ding et al., 2025; Ke et al., 2025) or applying MLLMs to natural image retrieval (Zhao et al., 2023; Zhang et al., 2025c). To the best of our knowledge, no existing work provides a systematic overview of the VDR

**Table 1:** Comparisons between relevant surveys & ours. We denote Retrieval as 🄡 and Understanding as 🅤.

| Survey | Venue | Scope | Setting | Technical Trends | | |
|---|---|---|---|---|---|---|
| | | | | LLM | RAG | Agent |
| Alaei et al. (2016) | IJCNN'16 | IR4Doc | 🄡 | | | |
| Zhou et al. (2017) | arxiv'17 | IR4Img | 🄡 | | | |
| Subramani et al. (2020) | NeurIPS Workshop'20 | DL4Doc | 🅤 | | | |
| Hameed et al. (2021) | CE'21 | IR4Img | 🄡 | | | |
| Cui et al. (2021) | ICDAR'21 | IR4Doc | 🅤 | | | |
| Sassioui et al. (2023) | WINCOM'23 | IR4Doc | 🅤 | | | |
| Zhao et al. (2023) | EMNLP Finding'23 | LLM4Img | 🄡 | ✔ | ✔ | |
| Ding et al. (2024) | arxiv'24 | DL4Doc | 🅤 | ✔ | | |
| Huang et al. (2024) | IEEE TKDE'24 | LLM4Doc | 🅤 | ✔ | | ✔ |
| Rombach & Fettke (2025) | ACM Computing Survey'24 | DL4Doc | 🅤 | ✔ | | |
| Zhang et al. (2025c) | arxiv'25 | LLM4Img | 🄡 | ✔ | | |
| Gao et al. (2025) | arxiv'25 | LLM4Doc | 🅤 | ✔ | ✔ | ✔ |
| Ding et al. (2025) | IJCAI Tutorial'25 | LLM4Doc | 🅤 | ✔ | | |
| Ke et al. (2025) | ACM TOIS'25 | LLM4Doc | 🅤 | ✔ | ✔ | ✔ |
| Zhang (2025) | AACL-IJCNLP'25 | LLM4Doc | 🄡 | ✔ | ✔ | |
| **Ours** | - | **LLM4Doc** | 🄡 | ✔ | ✔ | ✔ |

landscape through the specific lens of retrieval-focused methodologies in the age of LLMs, especially covering the emerging paradigms of RAG and Agent-based systems. Our survey aims to bridge this critical gap,

offering the **first comprehensive treatise on VDR that synthesizes foundational techniques with the latest breakthroughs driven by (M)LLMs**[2].

**Structure.** We begin from a **benchmark perspective**, systematically organizing the field by examining task formulations, basic settings, and dataset characteristics such as multilingual support and the growing emphasis on reasoning-intensive queries. Following this, we transition to a **methodology-centric analysis**, categorizing existing approaches into three primary paradigms: ❶ embedding models that serve as the foundation for retrieval, ❷ reranker models designed to refine initial retrieval results, and ❸ the increasingly prominent RAG pipelines and agentic systems. Finally, we conclude by discussing the challenges and outlining **future frontiers**, aiming to provide valuable insights and inspire subsequent research in the multimodal document intelligence community.

## 2 Benchmark Perspective

This section provides a systematic review of VDR evaluation landscape. We first establish a formal mathematical definition (▷ Section 2.1), and then analyze the current trends in terms of data scale and metrics (▷ Section 2.2), followed by a discussion on the emerging frontiers of multilingual support (▷ Section 2.3) and reasoning-intensive retrieval (▷ Section 2.4), which reflect the shift from keyword matching to complex document intelligence.

### 2.1 Formulation

VDR aims to identify the most relevant document images from a large-scale corpus based on a given query. Formally, let $\mathcal{C} = \{d_1, d_2, \ldots, d_N\}$ be a corpus of $N$ document pages, where each $d_j \in \mathcal{I}$ is a visually rich document image. Given a query $q$, the task is to produce a ranked list of documents from $\mathcal{C}$ such that the top-ranked items maximize a relevance score $s(q, d)$.

**Input and Modalities.** In the standard VDR setting, the query is typically a natural language string $q \in \mathcal{T}$ (text-to-image retrieval). However, the generalized VDR setting extends the query space to multimodal inputs, including images or interleaved text-and-image sequences $q \in \{\mathcal{T} \cup \mathcal{I}\}^*$. Unlike traditional OCR-based retrieval that treats documents as plain text, VDR models process the document $d$ directly in the visual domain, often representing it as a set of patch-level embeddings to preserve layout and graphical information.

**Mathematical Representation.** Following the late-interaction paradigm pioneered by ColPali (Faysse et al., 2024), a document page can be represented as a set of multiple patch-level embeddings $\mathbf{D} = \{\mathbf{d}_j\}_{j=1}^{N_p}$, where $\mathbf{d}_j \in \mathbb{R}^D$, and the query as a set of token-level embeddings $\mathbf{Q} = \{\mathbf{q}_i\}_{i=1}^{N_q}$. Relevance thus can be then computed via a late-interaction mechanism like `MaxSim` operation: $s(q, d) = \sum_{i=1}^{N_q} \max_{j=1}^{N_p} \mathbf{q}_i^\top \mathbf{d}_j$.

**Evaluation Metrics.** This section provides detailed mathematical formulations for the primary retrieval metrics used in the benchmarks discussed in this survey. For a given set of test queries $\mathcal{Q}$, the goal of a retrieval system is to return a ranked list of documents from a corpus $\mathcal{C}$ for each query $q \in \mathcal{Q}$.

❶ **Recall@k** Recall at k (Recall@k) measures the fraction of relevant documents that are successfully retrieved within the top-k results. It is a metric of coverage, evaluating how well the system is able to find all ground-truth documents. For a given query $q$, let $R_q$ be the set of ground-truth relevant documents and $L_k(q)$ be the ranked list of the top-$k$ retrieved documents. The overall Recall@k is the average score across all queries in $\mathcal{Q}$.

Recall@k is calculated as:

$$\text{Recall@}k = \frac{1}{|\mathcal{Q}|} \sum_{q \in \mathcal{Q}} \frac{|R_q \cap L_k(q)|}{|R_q|} \tag{1}$$

where $|\mathcal{Q}|$ is the total number of queries; $|R_q|$ is the total number of relevant documents for query $q$; $|R_q \cap L_k(q)|$ is the number of relevant documents found in the top-$k$ retrieved list.

---

[2]This version covers literature up to **1 April, 2026**, with updates scheduled every two months for the fast-growing research community.

❷ **Mean Reciprocal Rank (MRR)**  Mean Reciprocal Rank (MRR) evaluates the ranking of the *first* correct document retrieved. It is particularly useful for tasks where finding a single relevant item quickly is the primary goal. For each query $q$, the reciprocal rank is the inverse of the rank of the first relevant document. If no relevant document is retrieved, the reciprocal rank is 0.

MRR is calculated as:

$$\text{MRR} = \frac{1}{|\mathcal{Q}|} \sum_{q \in \mathcal{Q}} \frac{1}{\text{rank}_q} \tag{2}$$

where $\text{rank}_q$ is the position (rank) of the first relevant document for query $q$.

❸ **Mean Average Precision (mAP)**  Mean Average Precision (mAP) provides a comprehensive evaluation of a ranked list by considering both precision and recall. It rewards retrieving relevant documents at higher ranks. The Average Precision (AP) for a single query is first calculated by averaging the precision at each relevant document's position. mAP is then the mean of these AP scores over all queries.

The Average Precision for a single query $q$ is:

$$\text{AP}_q = \frac{1}{|R_q|} \sum_{k=1}^{|\mathcal{C}|} (P(k) \times \text{rel}(k)) \tag{3}$$

And mAP is the mean of all AP scores:

$$\text{mAP} = \frac{1}{|\mathcal{Q}|} \sum_{q \in \mathcal{Q}} \text{AP}_q \tag{4}$$

where $|\mathcal{C}|$ is the total number of documents in the corpus; $P(k)$ is the precision at rank $k$; $\text{rel}(k)$ is an indicator function that is 1 if the document at rank $k$ is relevant, and 0 otherwise.

❹ **Normalized Discounted Cumulative Gain (nDCG@k)**  Normalized Discounted Cumulative Gain (nDCG@k) is a measure of ranking quality that accounts for the graded relevance of documents. It assigns higher scores to more relevant documents placed at top ranks, with a logarithmic discount for documents at lower ranks. The score is normalized by the ideal ranking to fall between 0 and 1.

The DCG@k is first calculated as:

$$\text{DCG@}k = \sum_{i=1}^{k} \frac{\text{rel}_i}{\log_2(i+1)} \tag{5}$$

The nDCG@k is then:

$$\text{nDCG@}k = \frac{\text{DCG@}k}{\text{IDCG@}k} \tag{6}$$

where $k$ is the number of results being evaluated; $\text{rel}_i$ is the graded relevance score of the document at position $i$; IDCG@$k$ is the Ideal Discounted Cumulative Gain, representing the DCG score of the perfect ranking up to position $k$.

❺ **Hit Rate (HR@k)**  As used in benchmarks like `Double-Bench` (Shen et al., 2025), Hit Rate at k (HR@k) is a binary success metric that evaluates whether *at least one* relevant document is found within the top-k retrieved results. It is particularly useful for assessing whether the retrieval system can find any correct evidence, which is a prerequisite for downstream tasks. For multi-hop queries, a "hit" is only registered if evidence for *all* hops is found within the top-k results.

For a single-hop query $q$, the HR@k is:

$$\text{HR@}k(q) = \mathbb{I}(L_k(q) \cap R_q \neq \emptyset) \tag{7}$$

The overall HR@k is the average across all queries:

$$\text{HR@}k = \frac{1}{|\mathcal{Q}|} \sum_{q \in \mathcal{Q}} \text{HR@}k(q) \tag{8}$$

where $\mathbb{I}(\cdot)$ is the indicator function, which is 1 if the condition is true and 0 otherwise; $L_k(q)$ is the top-k retrieved list and $R_q$ is the set of ground-truth documents.

❻ **Averaged Normalized Longest Common Subsequence (ANLCS)**  Used in `VisDoMBench` (Suri et al., 2025) to evaluate evidence extraction, Averaged Normalized Longest Common Subsequence (ANLCS) measures the textual similarity between retrieved content and ground-truth evidence. Instead of evaluating ranking, it assesses the quality of the retrieved content itself. First, the Normalized Longest Common Subsequence (NLCS) is calculated for a pair of text strings, measuring their shared content normalized by their lengths.

The NLCS between a ground-truth evidence string $S_{gt}$ and a retrieved chunk string $S_{ret}$ is:

$$\text{NLCS}(S_{gt}, S_{ret}) = \frac{2 \times |\text{LCS}(S_{gt}, S_{ret})|}{|S_{gt}| + |S_{ret}|} \tag{9}$$

The ANLCS score for a query is then computed by finding the best-matching retrieved chunk for each ground-truth evidence chunk and averaging these scores. The final metric is the average over all queries.

$$\text{ANLCS@}k = \frac{1}{|\mathcal{Q}|} \sum_{q \in \mathcal{Q}} \left( \frac{1}{|G_q|} \sum_{g \in G_q} \max_{c \in C_k(q)} \text{NLCS}(g, c) \right) \tag{10}$$

where $\text{LCS}(S_{gt}, S_{ret})$ is the Longest Common Subsequence between the two strings; $|S|$ denotes the length of string $S$; $G_q$ is the set of ground-truth evidence texts for query $q$; $C_k(q)$ is the set of retrieved text chunks in the top-k results for query $q$.

## 2.2 Basic Setting

**Escalating Research Momentum.** Over the past two years, VDR has transitioned from a niche task to a central focus in both industry and academia. As shown in Table 2, the majority of specialized VDR benchmarks, such as ViDoRe seires (Faysse et al., 2024) and Real-MM-RAG (Wasserman et al., 2025b), emerged in 2024 and 2025. This surge is driven by the realization that (OCR-based) text-only retrieval fails to capture the visual nuances of documents, such as tables, charts, and spatial hierarchies (Zhang et al., 2025b; Most et al., 2025).

**Diverse Dataset Scales.** Current benchmarks exhibit a wide range of data magnitudes (Cao et al., 2025). While expert-annotated sets like SeaDoc (Xiao et al., 2025b) and M4DocBench (Dong et al., 2025c) focus on specialized samples, recent large-scale efforts have pushed boundaries. NL-DIR (Guo et al., 2025) provides 205k queries, and Jina-VDR (Macé et al., 2025) utilizes over 70k documents, reflecting a trend toward scaling both query volume and corpus diversity.

**Standardized Evaluation.** nDCG and Recall remain the primary metrics for performance measurement. However, as VDR is increasingly integrated into RAG, some benchmarks (*e.g.,* EVisRAG (Sun et al., 2025b) and MMLongBench-Doc (Ma et al., 2024b)) incorporate downstream Accuracy and F1 scores to evaluate how retrieval quality directly impacts final multimodal generation.

## 2.3 Multilingual Support

The majority of early VDR benchmarks are predominantly English-centric. However, recent work has begun to bridge this linguistic gap like multilingual text embedding benchmarks (Enevoldsen et al., 2025; Zhang et al., 2023). Jina-VDR (Macé et al., 2025) and Nayana-IR (Kolavi & Jain, 2025) represent a significant shift, supporting 20 and 22 languages respectively. Similarly, MIRACL-VISION (Osmulski et al., 2025) introduces a large-scale multilingual corpus covering 18 languages. This movement highlights the necessity for VDR models to move beyond English-specific representations and develop cross-lingual semantic alignment capabilities, particularly for visually rich documents.

| Category | Benchmark | Publication | | Dataset | | | | | Resource |
|---|---|---|---|---|---|---|---|---|---|
| | | Team | Venue | #Query* | #Corpus* | Multilingual | Reasoning | Retrieval Metric | |
| VDR | MRMR (Zhang et al., 2025e) | NTU | ICLR'26 | 1.50k | 42.80k | en | ✔ | nDCG / HR | |
| VDR | ViDoRe-V3 (Loison et al., 2026) | Illuin Tech | arxiv'26 | 3.10k | 26.00k | en, fr | - | nDCG | |
| VDR | IRPAPERS (Shorten et al., 2026) | Weaviate | arxiv'26 | 180 | 3.23k | en | - | Recall | |
| VDR | ViDoRe-V1 (Faysse et al., 2024) | Illuin Tech | ICLR'25 | 3.81k | 8.31k | en, fr | - | nDCG / Recall / MRR | |
| VDR | VisRAG (Yu et al., 2024) | Tsinghua | ICLR'25 | 3.61k | 20.64k | en | - | Recall / MRR | |
| VDR | SeaDoc (Xiao et al., 2025b) | Alibaba | NeurIPS'25 | 1.00k | 5.06k | 4 | - | nDCG | |
| VDR | MMDocRAG (Dong et al., 2025b) | Huawei | NeurIPS'25 | 4.06k | 14.87k | en | - | Recall | |
| VDR | Real-MM-RAG (Wasserman et al., 2025b) | IBM | ACL'25 | 4.55k | 8.60k | en | - | nDCG / Recall | |
| VDR | NL-DIR (Guo et al., 2025) | CAS | CVPR'25 | 205.00k | 41.80k | en | - | Recall / MRR | |
| VDR | OpenDocVQA (Tanaka et al., 2025) | NTT | CVPR'25 | 43.47k | 206.27k | en | - | nDCG | |
| VDR | ViDoSeek (Wang et al., 2025b) | Alibaba | EMNLP'25 | 1.14k | 5.39k | en | - | Recall / MRR | |
| VDR | MMDocIR (Dong et al., 2025a) | Huawei | EMNLP'25 | 1.66k | 20.40k | en | - | Recall | |
| VDR | Jina-VDR (Günther et al., 2025) | Jina AI | EMNLP'25 | 57.33k | 70.62k | 20 | - | nDCG | |
| VDR | VisDoMBench (Suri et al., 2025) | UMD | NAACL'25 | 2.27k | 8.30k | en | - | ANLCS | |
| VDR | ViDoRe-V2 (Macé et al., 2025) | Illuin Tech | arxiv'25 | 1.15k | 4.54k | en, fr | - | nDCG | |
| VDR | EVisRAG (Sun et al., 2025b) | PKU | arxiv'25 | 4.26k | - | en | - | Acc / F1 | |
| VDR | Double-Bench (Shen et al., 2025) | SCUT | arxiv'25 | 5.17k | 73.06k | 6 | - | HR | |
| VDR | VisR-Bench (Chen et al., 2025c) | UB | arxiv'25 | 35.57k | 35.63k | 16 | - | Acc | |
| VDR | MR2-Bench (Zhou et al., 2025a) | BAAI | arxiv'25 | 1.31k | 47.74k | en | ✔ | nDCG / Recall | |
| VDR | MIRACL-VISION (Osmulski et al., 2025) | NVIDIA | arxiv'25 | 7.90k | 338.73k | 18 | - | nDCG | |
| VDR | UniDoc-Bench (Peng et al., 2025) | Salesforce | arxiv'25 | 1.74k | 70.00k | en | - | Recall / Precision | |
| VDR | ViMDoc (Kim et al., 2025a) | KAIST | arxiv'25 | 10.90k | 76.35k | en | - | Recall | |
| VDR | M4DocBench (Dong et al., 2025c) | Huawei | arxiv'25 | 158 | 6.18k | en, zh | ✔ | Recall | |
| VDR | SDS KoPub VDR (Lee et al., 2025a) | Samsung | arxiv'25 | 600 | 40.78k | en, ko | - | nDCG / Recall | |
| VDR | Nayana-IR (Kolavi & Jain, 2025) | CognitiveLab | arxiv'25 | 1.00k | 5.40k | 22 | - | nDCG / Recall / mAP / MRR | |
| VDR | MMLongBench-Doc (Ma et al., 2024b) | NTU | NeurIPS'24 | 1.08k | 6.41k | en | - | Acc / F1 | |
| Gen. | MMEB-V2 (Meng et al., 2025) | Salesforce | TMLR'26 | - | - | en | - | nDCG | |
| Gen. | MM-BRIGHT (Abdallah et al., 2026) | University of Innsbruck | arxiv'26 | 2.80k | 2.50M | en | ✓ | nDCG | |
| Gen. | ARK-Bench (Lin et al., 2026) | SCU | arxiv'26 | 1.55k | 36.03k | en | ✓ | nDCG / Recall | |
| Gen. | MMEB (Jiang et al., 2024c) | Waterloo & Salesforce | ICLR'25 | 12.00k | 12.00M | en | - | Precision | |
| Gen. | Visual Haystacks (Wu et al., 2024b) | UCB | ICLR'25 | 20.10k | 97.00k | en | - | Acc | |
| Gen. | COCO-Facet (Li et al., 2025d) | Washington | NeurIPS'25 | 9.11k | 911.20k | en | - | Recall | |
| Gen. | UMRB (Zhang et al., 2024c) | PolyU & Alibaba | CVPR'25 | 200.00k | 40.00M | en | - | nDCG / Recall | |
| Gen. | MIEB (Xiao et al., 2025c) | Durham University | ICCV'25 | 1.25M | 7.62M | 38 | - | nDCG / Recall / mAP | |
| Gen. | MMNeedle (Wang et al., 2025a) | Rutgers University | NAACL'25 | 280.00k | 40.00k | en | - | Acc | |
| Gen. | MMMEB (Musacchio et al., 2025) | Uniba | arxiv'25 | 9.20k | 9.20k | 5 | - | Precision | |
| Gen. | MM-NIAH (Wang et al., 2024b) | FDU & Shanghai AI Lab | NeurIPS'24 | 5.58k | - | en | - | Acc | |
| Gen. | M2KR (Lin et al., 2024b) | Cambridge | ACL'24 | 39.79k | 514.85k | en | - | Recall | |
| Gen. | M-BEIR (Wei et al., 2024a) | Waterloo | ECCV'24 | 190k | 5.6M | en | - | Recall | |

**Table 2:** Comparison of VDR and general multimodal retrieval benchmarks. In *Resource* column, we denote the corresponding github codebase/huggingface/paper(*e.g.,* arxiv paper, technical report or blog) as 🐙 / 🤗 / 📄. * indicates that #query and #corpus corresponds to retrieval-related #valuation sample and #candidate, respectively, for general multimodal retrieval benchmarks.

## 2.4 Reasoning-Intensive Setting

The rapid advancement of MLLMs has fundamentally shifted the objective of information retrieval from surface-level semantic matching to deep cognitive reasoning (Bi et al., 2025; Su et al., 2025b; Lin et al., 2025b; Yan et al., 2024). While **text-centric retrieval has already pioneered "reasoning-aware" capabilities** through benchmarks[3] (*e.g.,* BRIGHT (Su et al., 2024), RAR-b (Xiao et al., 2024), ATEB (Han et al., 2025a), R2MED (Li et al., 2025b)), embedding models (*e.g.,* ReasonIR (Shao et al., 2025), ReasonEmbed (Chen et al., 2025d), RaDeR (Das et al., 2025), DIVER-Retriever (Long et al., 2025), RITE (Liu et al., 2025f), REAPER (Joshi et al., 2024)) and reranker models (*e.g.,* Reason-to-Rank (Ji et al., 2024), InteRank (Samarinas & Zamani, 2025), Rank1 (Weller et al., 2025b), Rank-K (Yang et al., 2025a), TFRank (Fan et al., 2025), TS-SetRank (Huang et al., 2025a), InsertRank (Seetharaman et al., 2025), RGS (Xu & Chen, 2025)), VDR is currently witnessing a pivotal transition. In VDR, the density of textual information combined with intricate visual layouts (*e.g.,* charts, tables, and diagrams) necessitates a level of logical deduction that extends far beyond traditional OCR-based matching (Duan et al., 2025; Liao et al., 2025; Nacson et al., 2025; Jiang et al., 2025).

Recent pioneering efforts have begun to formalize this reasoning-intensive frontier through three distinct perspectives:

❶ **Vision-Centric Logic and Abstract Reasoning.** As highlighted by **MR2-Bench** (Zhou et al., 2025a), current VDR models often suffer from "shallow matching," where they succeed by identifying object-text correlations but fail at logical, spatial, or causal inference. MR2-Bench introduces vision-centric tasks such as *Visual Puzzle* (solving Raven-style matrices) and *Visual Illustration Search* (*e.g.,* matching a mathematical

---

[3]Instruction-following retrieval (*e.g.,* FollowIR (Weller et al., 2024) and mFollowIR (Weller et al., 2025a)), long-context retrieval (*e.g.,* LongEmbed (Zhu et al., 2024a)) and code retrieval benchmarks (*e.g.,* CoIR (Li et al., 2024b) and CodeRAG-Bench (Wang et al., 2024d)) are not within the scope of reasoning-intensive retrieval in this survey.

formula to its corresponding geometric proof). Their findings reveal a massive "reasoning gap": models achieving high scores on general multimodal benchmarks (*e.g.,* MMEB) exhibit a significant performance drop when required to solve abstract visual analogies or multi-image relational scenarios.

❷ **Expert-Level Multidisciplinary and Contradiction Reasoning.** Moving beyond general knowledge, **MRMR** (Zhang et al., 2025e) introduces *Contradiction Retrieval*, a novel task requiring models to identify rules or requirements that conflict with a given case description (*e.g.,* identifying a traffic violation in a visual scene based on a text-based rulebook). MRMR focuses on expert domains like medicine and engineering, where visually rich documents (*e.g.,* pathological slides) require specialized interpretation. Their study underscores that text-only retrievers augmented with high-quality captions often outperform native multimodal models, suggesting that superior logical deduction in LLMs still outweighs current visual-semantic alignment in native multimodal embedders.

❸ **Agentic, Multi-hop, and Process-Oriented Reasoning.** In complex enterprise scenarios, **M4DocBench** (Dong et al., 2025c) redefines reasoning as an iterative, agentic process. It introduces the *M4 framework* (Multi-modal, Multi-hop, Multi-document, and Multi-turn), focusing on deep research tasks where evidence is scattered across dozens of documents. Unlike single-shot retrieval, M4DocBench evaluates a system's ability to perform *Query Decomposition* and iterative refinement. It underscores that deep document intelligence requires "Strategic Planning"—the ability to filter relevant documents from noisy collections and adaptively select the optimal retrieval granularity (*e.g.,* chunk, page, or summary) based on the evolving state of the research workflow.

**Future Frontiers.** Despite these early triumphs, several reasoning challenges remain largely unexplored. Future VDR research must address **implicit retrieval intents**, where queries involve fuzzy constraints that can only be resolved through world-knowledge synthesis (Zhang et al., 2025g; Wei et al., 2024b). Furthermore, the development of **active retrieval agents**—capable of self-correcting their search path when initial reasoning leads to dead ends—will be paramount to unlocking the full potential of multimodal document intelligence in open-domain, large-scale scenarios (Zhu et al., 2024b; Liu et al., 2025c; Zhang et al., 2025d).

## 3    Methodology Perspective

This section deconstructs the technical trends that underpin modern VDR. We begin by examining embedding models (▷ Section 3.1), as summarized in Table 3. Next, we analyze reranker models (▷ Section 3.2), as shown in Table 4. Finally, we discuss how they are integrated into sophisticated RAG and Agentic systems (▷ Section 3.3).

### 3.1    Embedding Models

#### 3.1.1    Formulation

In the context of VDR, an embedding model, denoted by an encoder function $E(\cdot)$, processes multimodal inputs to produce vector representations. For a given query $q$, typically tokenized into a sequence $Q = \{t_1^q, t_2^q, \ldots, t_M^q\}$, and a document page $d$, represented as a sequence of image patches $D = \{p_1^d, p_2^d, \ldots, p_N^d\}$, the objective is to learn an encoder that aligns their representations. While early models produced a single vector for each item (Zhang et al., 2024c; Liu et al., 2025e; Jiang et al., 2024c), recent VDR models adopt a multi-vector paradigm. The relevance score $s(q, d)$ between the query embedding set $E(Q) = \{\mathbf{e}_{t_i}^q\}_{i=1}^M$ and the document patch embedding set $E(D) = \{\mathbf{e}_{p_j}^d\}_{j=1}^N$ is often computed using a late-interaction mechanism like MaxSim (Faysse et al., 2024).

#### 3.1.2    Common VDR Training Sets.

The primary distinction between specialized VDR models and general multimodal embedding models lies in their training data. VDR models are fine-tuned extensively on visual document datasets to master the fine-grained semantic and layout-aware understanding required for this domain. We discuss common VDR training sets as follows.

| Category | Embedding | Publication | | Model | | | | | | Novelty | Resource |
|---|---|---|---|---|---|---|---|---|---|---|---|
| | | Team | Venue | Para. | Backbone | Multilingual | Multi-Vec | Training-Free | Generative | | |
| VDR | Nemotron ColEmbed V2 (Moreira et al., 2026) | NVIDIA | ECIR'26 | 4B/8B | Qwen3-VL/Llama3.2 | en | ✓ | - | - | M D | hf, pdf |
| VDR | NanoVDR (Liu et al., 2026b) | Aalto University | arxiv'26 | 69M/112M/151M | BERT | en | - | - | - | T E | hf |
| VDR | CausalEmbed (Huo et al., 2026) | HKUST(GZ) & Alibaba | arxiv'26 | 3B | Qwen2.5-VL/PaliGemma | en | ✓ | - | ✓ | T E | github, hf, pdf |
| VDR | ColParse (Yan et al., 2026c) | HKUST(GZ) & Alibaba | arxiv'26 | 2B/7B | Qwen2/2.5-VL | en | ✓ | ✓ | - | M E | pdf |
| VDR | ColChunk (Yan et al., 2026a) | HKUST(GZ) & Alibaba | arxiv'26 | 2B/7B | Qwen2/2.5-VL | en | ✓ | ✓ | - | M E | pdf |
| VDR | Ops-Colqwen3 (OpenSearch-AI, 2026) | Alibaba | blog'26 | 4B | Qwen3-VL | 30 | ✓ | - | - | M D | hf |
| VDR | ColPali/ColQwen (Faysse et al., 2024) | Illuin Tech | ICLR'25 | 3B | PaliGemma | en | ✓ | - | - | M D T | github, hf, pdf |
| VDR | Unveil (Sun et al., 2025a) | PKU | ACL'25 | 3B | MiniCPM-V | en | - | - | - | M | pdf |
| VDR | ColMate (Masry et al., 2025) | ServiceNow | EMNLP'25 | 3B | PaliGemma | en/fr | ✓ | - | - | M | hf, pdf |
| VDR | ColFlor (Masry & Hoque, 2025) | York Uni. | arxiv'25 | 0.2B | Florence-2 | en/fr | ✓ | - | - | E | github, hf |
| VDR | ColModernVBERT (Teiletche et al., 2025) | Illuin Tech | arxiv'25 | 0.3B | SigLIP | en | ✓ | - | - | M E | hf, pdf |
| VDR | Nemoretriever ColEmbed (Xu et al., 2025a) | NVIDIA | arxiv'25 | 1B/3B | Llama-3.2 | 18 | ✓ | - | - | M | hf, pdf |
| VDR | jina-embeddings-v4 (Günther et al., 2025) | Jina AI | arxiv'25 | 4B | Qwen2.5-VL | 20 | ✓ | - | - | M T | hf, pdf |
| VDR | Tomoro-ColQwen (Huang & Tan, 2025) | Tomoro AI | arxiv'25 | 4B/8B | Qwen2.5/3 | en/fr | ✓ | - | - | E | hf, pdf |
| VDR | ColNetraEmbed (Kolavi & Jain, 2025) | CognitiveLab | arxiv'25 | 4B | Gemma 3 | 22 | ✓ | - | - | D | hf, pdf |
| VDR | Granite-vision-embedding (Team et al., 2025a) | IBM | arxiv'25 | 2B | Granite-3.1 | en | - | - | - | D E | hf, pdf |
| VDR | SauerkrautLM-ColQwen (Golchinfar, 2025) | VAGO Solutions | blog'25 | 2B/4B/8B | Qwen3-VL | 6 | ✓ | - | - | D | hf |
| VDR | vdr-2b-multi-v1 (LlamaIndex, 2025) | LlamaIndex | blog'25 | 2B | Qwen2VL | 5 | - | - | - | D E | hf |
| VDR | Eager Embed (Balarini, 2025) | Eagerworks | blog'25 | 4B | Qwen2.5-VL | en | - | - | - | - | github, hf |
| VDR | ColNomic Embed Multimodal (Team, 2025) | Nomic AI | blog'25 | 3B/7B | Qwen2.5-VL | 5 | ✓ | - | - | D T | hf, pdf |
| VDR | DSE (Ma et al., 2024a) | Waterloo | EMNLP'24 | 4B | Phi-3-Vision | en | - | - | - | D | pdf |
| Gen. | MetaEmbed (Xiao et al., 2025d) | Meta | ICLR'26 | 3-32B | Qwen2.5/Llama-3.2 | en/fr | ✓ | - | - | M E | pdf |
| Gen. | UME-R1 (Lan et al., 2025b) | Tencent | ICLR'26 | 2B/7B | Qwen2-VL | en | - | - | ✓ | M T | github, hf, paper |
| Gen. | TTE (Cui et al., 2025b) | Meta | ICLR'26 | 2B/7B | Qwen2-VL | en/fr | - | - | ✓ | M | |
| Gen. | U-MARVEL (Li et al., 2025e) | NJU | ICLR'26 | 7B | Qwen2-VL | en | - | - | - | M T | github, hf, paper |
| Gen. | VLM2Vec-V2 (Meng et al., 2025) | Salesforce | TMLR'26 | 2B | Qwen2-VL | en/fr | - | - | - | D | github, hf, paper |
| Gen. | Evo-Retriever (Li et al., 2026c) | Alibaba | CVPR'26 | 3B/8B | Qwen2.5-VL | en | ✓ | - | - | T | github, hf |
| Gen. | MuCo (Gu et al., 2026) | NAVER AI | CVPR'26 | 2B/7B | Qwen2-VL | en | - | - | - | T | github |
| Gen. | UniME-V2 (Gu et al., 2025b) | MiroMind | AAAI'26 | 2B/7B | Qwen2-VL/LLaVA | en | - | - | - | D T | github, hf, paper |
| Gen. | Qwen3-VL-Embedding (Li et al., 2026a) | Alibaba | arxiv'26 | 2B/8B | Qwen3-VL | 30 | - | - | - | D T | github, hf, paper |
| Gen. | V-Retrver (Chen et al., 2026a) | Tsinghua | arxiv'26 | 7B | Qwen2.5-VL | en | - | - | ✓ | M T | github, paper |
| Gen. | TTE-v2 (Cui et al., 2025a) | Meta | arxiv'26 | 2B/7B | Qwen2-VL | en | - | - | ✓ | M | paper |
| Gen. | Embed-RL (Jiang et al., 2026) | Tsinghua & Kuaishou | arxiv'26 | 2B/4B | Qwen3-VL | en | - | - | ✓ | M T | github, hf, paper |
| Gen. | PLUME (He et al., 2026) | SEU & CAS | arxiv'26 | 2B | Qwen2-VL | en | - | - | ✓ | M | github, paper |
| Gen. | RAR (Zhang et al., 2026a) | Wisconsin-Madison | arxiv'26 | 8B | Qwen3-VL | 30 | - | - | ✓ | M D | github, paper |
| Gen. | TRACE (Hao et al., 2026) | CAS | arxiv'26 | 7B | Qwen2.5-VL | en | - | - | ✓ | M D | paper |
| Gen. | MMEmb-R1 (Wang et al., 2026) | CUHK & ByteDance | arxiv'26 | 2B/4B/7B | Qwen2/3-VL | en | - | - | ✓ | M T | github, paper |
| Gen. | Magic-MM-Embedding (Li et al., 2026b) | Huawei | arxiv'26 | 7B | InternVL3 | en | - | - | - | M T E | github, paper |
| Gen. | e5-omni (Chen et al., 2026b) | RUC | arxiv'26 | 3B/8B | Qwen2.5-Omni | en | - | - | - | M T | hf, pdf |
| Gen. | AGFF-EMBED (Hu et al., 2026) | Tencent | arxiv'26 | 7B | Qwen2-VL | en | - | - | - | M T | pdf |
| Gen. | CoMa (Li et al., 2025a) | CAS | arxiv'26 | 3B/7B | Qwen2.5-VL | en | - | - | - | M E | |
| Gen. | MM-Embed (Lin et al., 2024a) | NVIDIA | ICLR'25 | 7B | LLaVA-NeXT | en | - | - | - | T | hf, pdf |
| Gen. | VLM2Vec (Jiang et al., 2024c) | Waterloo & Salesforce | ICLR'25 | 8B | Phi-3.5-V | en | - | - | - | M D | github, hf, pdf |
| Gen. | B3 (Thirukovalluru et al., 2025) | Duke | NeurIPS'25 | 2B/7B/8B | Qwen2-VL/InternVL3 | en | - | - | - | T E | github, hf, paper |
| Gen. | LCO-Embedding (Xiao et al., 2025b) | Alibaba | NeurIPS'25 | 3B/7B | LLaVA/Qwen2.5-VL | 38 | - | - | - | M | github, hf, paper |
| Gen. | Retrv-R1 (Zhu et al., 2025b) | CityU HK | NeurIPS'25 | 3B/7B | Qwen2.5-VL | en | - | - | ✓ | M D T | paper |
| Gen. | MMRet-MLLM (Zhou et al., 2025b) | BUPT | ACL'25 | 7B | LLaVA-1.6 | en | - | - | - | D | github, hf, paper |
| Gen. | UniSE (Liu et al., 2025h) | BAAI | ACL'25 | 0.4/2B | CLIP/Qwen2-VL | en | - | - | - | M D | github, hf, paper |
| Gen. | GME (Zhang et al., 2024c) | PolyU & Alibaba | CVPR'25 | 7B | Qwen2-VL | en | - | - | - | D | hf |
| Gen. | LamRA (Liu et al., 2025e) | SJTU & Xiaohongshu | CVPR'25 | 7B | Qwen2-VL | en | - | - | - | M | github, hf, paper |
| Gen. | VladVA (Ouali et al., 2025) | Samsung | CVPR'25 | 7B | LLaVA-1.5 | en | - | - | ✓ | M | paper |
| Gen. | CAFe (Yu et al., 2025a) | Meta | ICCV'25 | 0.5B/7B | LLaVA-OV | en | - | - | ✓ | T | github, paper |
| Gen. | UniME-V1 (Gu et al., 2025a) | Sydney | MM'25 | 4B/7B | Phi-3.5/LLaVA | en | - | - | - | T | github, hf, paper |
| Gen. | CoMa (Li et al., 2025a) | Kuaishou | arxiv'25 | 3B/7B | Qwen2.5-VL | en | - | - | - | M E | paper |
| Gen. | FreeRet (Zhu et al., 2025c) | NJU | arxiv'25 | 2-32B | Qwen2/2.5-VL/InternVL3 | en | - | ✓ | - | M | paper |
| Gen. | LLaVE (Lan et al., 2025a) | Tencent | arxiv'25 | 0.5B/7B | LLaVA/Aquila | en | - | - | - | T | github, hf, paper |
| Gen. | mmE5 (Chen et al., 2025b) | RUC | arxiv'25 | 11B | Llama-3.2 | 93 | - | - | - | D | github, hf, paper |
| Gen. | MoCa (Chen et al., 2025a) | RUC | arxiv'25 | 3B/7B | Qwen2.5-VL | en/fr | - | - | - | M T | github, hf, paper |
| Gen. | PDF (Wang et al., 2025f) | Alibaba | arxiv'25 | 2B/7B | LLaVA/Qwen2-VL | en | - | - | - | M T | github |
| Gen. | QQMM (Xue et al., 2025) | Tencent | arxiv'25 | 7B | LLaVA/Qwen2-VL | en | - | - | - | T | github |
| Gen. | ReMatch (Liu et al., 2025d) | Xiaohongshu | arxiv'25 | 2B/7B | Qwen2/2.5-VL | en | - | - | - | M | github |
| Gen. | RGE (Liu et al., 2025a) | SenseTime | arxiv'25 | 3B | Qwen2.5-VL | en | - | - | ✓ | M T | github, hf, paper |
| Gen. | RzenEmbed (Jian et al., 2025) | 360 AI | arxiv'25 | 2B/8B | Qwen2-VL | en | - | - | - | D T | hf, pdf |
| Gen. | Unite (Kong et al., 2025) | Kuaishou | arxiv'25 | 2B/7B | Qwen2-VL | en | - | - | - | D T | github, hf, paper |
| Gen. | VIRTUE (Wang et al., 2025e) | Sony | arxiv'25 | 2B/7B | Qwen2-VL | en | - | - | - | M D | paper |
| Gen. | xVLM2Vec (Musacchio et al., 2025) | UBAM | arxiv'25 | 4B | Phi-3.5-V | 5 | - | - | - | D D | github, hf, paper |
| Gen. | SAIL-Embedding (Lin et al., 2025a) | ByteDance | arxiv'25 | - | SAIL-VL | en | - | - | - | D T | paper |
| Gen. | Ops-MM-embedding (Alibaba, 2025b) | Alibaba | blog'25 | 2B/7B | Qwen2-VL | en | - | - | - | D | hf |
| Gen. | EvoQwen2.5-VL-Retriever (Alibaba, 2025a) | Alibaba | blog'25 | 3B/7B | Qwen2.5-VL | en | ✓ | - | - | D | hf |
| Gen. | Seed1.6-Embedding (ByteDance, 2025) | ByteDance | blog'25 | - | Seed1.6-flash | en | - | - | - | D | pdf |
| Gen. | PreFLMR (Lin et al., 2024b) | Cambridge | ACL'24 | 2B | BERT | en | ✓ | - | - | M D | github, hf, paper |
| Gen. | Vista (Zhou et al., 2024) | BUPT | ACL'24 | 0.2B | BGE-v1.5 | en | - | - | - | M D | github, hf, pdf |
| Gen. | UniIR (Wei et al., 2024a) | Waterloo | ECCV'24 | 0.4B | CLIP | en | - | - | - | D | github, hf, paper |
| Gen. | M-Solomon (Kim et al., 2025b) | NC AI | CIKM'24 | 7B | Qwen2-VL | en | - | - | - | M | paper |
| Gen. | Nomic Embed Vision (Nussbaum et al., 2024) | Nomic AI | arxiv'24 | 0.2B | BERT | en | - | - | - | - | hf, pdf |
| Gen. | E5-V (Jiang et al., 2024b) | Beihang | arxiv'24 | 8B | LLaVA-NeXT | en | - | - | ✓ | M | github, paper |

**Table 3:** Comparison of VDR and general multimodal embedding models. In *Novelty* column, we denote Model-/Data-/Training-/Efficiency-level contribution as **M** / **D** / **T** / **E**. In *Resource* column, we denote the corresponding github codebase/huggingface/paper(*e.g.,* arxiv paper, technical report or blog) as 🐙 / 🤗 / 📄.

❶ **colpali-train-set.** This dataset serves as the training data for the ColPali model[4] and represents a hybrid approach to data collection (Faysse et al., 2024). It comprises approximately 127,000 query-image pairs, strategically combining established academic benchmarks (63%), such as DocVQA (Mathew et al., 2021), InfoVQA (Mathew et al., 2022), TAT-DQA (Zhu et al., 2022) and arXivQA (Li et al., 2024a), with a custom synthetic dataset (37%). The synthetic component was created from a diverse collection of web-crawled PDF documents, with pseudo-questions generated by Claude-3 Sonnet. The dataset is intentionally English-only, designed to facilitate research into zero-shot cross-lingual generalization capabilities of VDR models.

---

[4]https://huggingface.co/datasets/vidore/colpali_train_set

❷ **VisRAG-Ret-Train-Synthetic-data.** As the synthetic training component for the VisRAG model[5] (Yu et al., 2024; Sun et al., 2025b), this dataset is composed entirely of VLM-generated data, totaling around 239k query-document pairs. The corpus was constructed from web-crawled PDFs spanning diverse domains, including college-level textbooks[6], academic papers from premier conferences like ICML'23 and NeurIPS'23, and product manuals[7]. A powerful VLM, GPT-4o, was leveraged to generate pseudo-queries for these document pages, creating a large-scale resource tailored for training retrieval models on a variety of document layouts and topics.

❸ **vdr-multilingual-train.** This dataset[8] marks a significant step towards multilingual VDR, containing nearly 500,000 query-image samples across five languages (English, Spanish, Italian, German, and French). Its construction involved a highly sophisticated pipeline. First, a diverse corpus of ~50k documents was scraped from the internet using topic-based search queries for each language. A key innovation was the use of layout analysis to sample pages, ensuring an even distribution of text-only, visual-only, and mixed-modality pages. Synthetic queries were then generated using powerful VLMs (Gemini-1.5-Pro and Qwen2-VL-72B) with an advanced prompting technique that distinguished between general and specific questions to improve query quality. The dataset underwent rigorous cleaning, filtering, and hard-negative mining to enhance its utility for training robust retrieval models.

❹ **VDR_MEGA_MultiDomain_DocRetrieval.** This dataset[9] represents the largest and most comprehensive resource to date, amalgamating approximately 1.09 million examples across five languages. It functions as a meta-dataset, strategically fusing the three previously mentioned datasets (`colpali-train-set`, `VisRAG-Ret-Train-Synthetic-data`, and `vdr-multilingual-train`) with several new, domain-specific collections. These additions cover specialized fields such as military[10], energy[11], hydrogen technology[12], and geotechnical engineering[13]. By unifying multiple datasets, it provides unparalleled scale and diversity in both language and subject matter, making it an ideal resource for training highly generalized and robust VDR systems.

❺ **docmatix.** This is an interleaved multimodal pre-training dataset[14] created for the modality-aware continual pre-training of MoCa models (Chen et al., 2025a). It is adapted from the original Docmatix dataset (Laurençon et al., 2024), a massive-scale Document Visual Question Answering resource containing approximately 2.4 million images and 9.5 million question-answer pairs, which was initially used to fine-tune the Idefics3 model. The adaptation process involves transforming the original question-answer pairs by concatenating document screenshots with their corresponding texts, thereby creating an interleaved format suitable for continuous pre-training.

**Common Characteristics.** The current generation of VDR training sets reveals several unifying trends. Firstly, there is a clear paradigm shift towards **leveraging synthetic data generation at scale**. All major datasets heavily rely on powerful VLMs to create pseudo-queries for large, unannotated document corpora, effectively bypassing the bottleneck of manual annotation. Secondly, these corpora are primarily built from **web-crawled PDF documents**, which provides a rich diversity of layouts, domains, and styles that mirror real-world scenarios. Thirdly, there is a growing emphasis on **sophisticated data curation**, including techniques like layout-aware page sampling and automated hard-negative mining, to improve training efficiency and model performance. Finally, a clear trajectory exists towards **increasing scale and multilingualism**, evolving from English-only datasets to massive, multi-language compilations that enable the development of globally competent models.

---

[5]https://huggingface.co/datasets/openbmb/VisRAG-Ret-Train-Synthetic-data
[6]https://openstax.org/
[7]https://www.manualslib.com/
[8]https://huggingface.co/datasets/llamaindex/vdr-multilingual-train
[9]https://huggingface.co/datasets/racineai/VDR_MEGA_MultiDomain_DocRetrieval
[10]https://huggingface.co/datasets/racineai/VDR_Military
[11]https://huggingface.co/datasets/racineai/VDR_Energy
[12]https://huggingface.co/datasets/racineai/VDR_Hydrogen
[13]https://huggingface.co/datasets/racineai/VDR_Geotechnie
[14]https://huggingface.co/datasets/moca-embed/docmatix

**Future Directions.** Looking ahead, the optimization of VDR training sets can be advanced in several key directions. The most critical frontier is moving **beyond simple semantic matching towards reasoning-intensive data**. Future datasets should include query-document pairs that necessitate multi-hop, logical, or causal reasoning to find the correct answer, mirroring the challenges posed by benchmarks like MRMR and M4DocBench. Secondly, there is a need for **enhanced data authenticity and complexity**. While VLM-generated queries are scalable, they can lack the nuance and "messiness" of real user intent. Future work could explore mining queries from anonymized user logs or using agentic workflows to simulate more realistic information-seeking behaviors. Lastly, training sets could benefit from **finer-grained structural annotations**. Instead of just matching a query to a page, future datasets could provide explicit links to specific sub-page elements (*e.g.,* a single row in a table, a data point in a chart, or a specific paragraph), which would be invaluable for training models that can perform precise, element-level evidence retrieval.

### 3.1.3 Technical Trends

**Model Choices.** The field has witnessed a clear trend in both model scale and architecture. While early models were based on smaller backbones like BERT (Nussbaum et al., 2024), current VDR embeddings are predominantly built on powerful MLLMs, such as PaliGemma (Beyer et al., 2024) and Qwen-VL series (Bai et al., 2025). parameter counts also have escalated, with SOTA models typically ranging from 2 to 8 billion parameters, a significant increase that enables more nuanced comprehension of complex document structures.

**Multilingual Support.** As document intelligence applications become increasingly global, multilingual capability has emerged as a critical focus in VDR. Unlike general-domain image retrieval where this aspect is less emphasized, the prevalence of multilingual business reports, academic papers, and official forms necessitates cross-lingual understanding. Recent models like jina-embeddings-v4 (Günther et al., 2025) and Nemoretriever (Xu et al., 2025a) reflect this shift, offering support for over 20 languages and demonstrating strong performance on multilingual VDR benchmarks such as Jina-VDR (Günther et al., 2025) and Nayana-IR (Kolavi & Jain, 2025).

**Multi-Vector Representation.** The multi-vector paradigm, popularized in the VDR domain by ColPali (Faysse et al., 2024), has become a dominant approach for fine-grained retrieval (Li et al., 2025c; Yang et al., 2026; Chaffin et al., 2026; Zhu et al., 2026), as shown in Figure 3. Instead of compressing an entire document or query into a single embedding, this method generates a set of token-level or patch-level embeddings. This granular representation is particularly effective for VDR because it enables "late interaction" matching

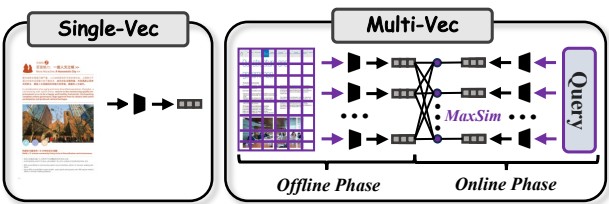

**Figure 3:** Single-vec (*left*) and multi-vec (*right*) VDR.

(Khattab & Zaharia, 2020), where specific phrases in a query can be precisely aligned with corresponding visual or textual regions in a document, thus preserving local details often lost in single-vector representations. ColPali (Faysse et al., 2024) adapts PaliGemma to produce multi-vector outputs by treating the embeddings of individual image patches as distinct vectors for late interaction. PreFLMR (Lin et al., 2024b) generates its multi-vector representations by concatenating embeddings from text tokens with both global and patch-level visual features, which are further refined through cross-attention to be query-aware. Addressing efficiency, ColModernVBERT (Teiletche et al., 2025) demonstrates that a compact, bidirectional language encoder can be effectively aligned with a vision encoder to generate granular embeddings; while MetaEmbed (Xiao et al., 2025d) introduces a fixed set of learnable "Meta Tokens" whose final hidden states serve as a compact and scalable multi-vector representation, enabling flexible trade-offs between retrieval quality and efficiency at test-time. More recently, ColParse (Yan et al., 2026c) introduces a layout-informed paradigm that leverages document parsing models to extract a compact set of semantically meaningful sub-images. By fusing these local layout-aware embeddings with a global page-level vector, ColParse achieves superior retrieval performance while significantly reducing storage requirements. Meanwhile, CausalEmbed (Huo et al., 2026) reframes the problem through an auto-regressive generative lens. It treats multi-vector embedding

construction as a sequential latent space generation task, enabling flexible test-time scaling, where retrieval precision can be dynamically adjusted by modifying the token budget at inference, thereby achieving high-fidelity representations with only a fraction of the conventional token count.

**Training Paradigm Exploration.** The predominant training method for VDR embedding models is end-to-end supervised fine-tuning using a contrastive loss, such as InfoNCE (Oord et al., 2018). This objective pulls positive query-document pairs closer together in the embedding space while pushing negative pairs apart. Beyond this standard approach, two novel paradigms are gaining traction.

The first paradigm explores **training-free methods**, which leverage the inherent knowledge of pre-trained MLLMs. E5-V (Jiang et al., 2024b) pioneers this by using carefully designed prompts to elicit universal embeddings directly from the MLLM's vocabulary space. FreeRet (Zhu et al., 2025c) introduces a plug-and-play framework that uses off-the-shelf MLLMs for both embedding-based search and multiple-choice question based reranking, all without any parameter updates.

The second paradigm explores how to harness the **generative capabilities** of MLLMs to enhance retrieval. Early works such as CAFe (Yu et al., 2025a) and VladVA (Ouali et al., 2025) introduced hybrid training frameworks; they jointly optimize a contrastive loss for discriminative power with an autoregressive, next-token prediction loss to preserve and leverage the model's inherent generative abilities. More recent approaches explicitly integrate reasoning as a precursor to embedding. For example, RGE (Liu et al., 2025a) and TTE (Cui et al., 2025b) both propose a "think-then-embed" process, where the model first generates an explicit rationale or reasoning trace, and the final embedding is conditioned on this generated context to capture more nuanced semantics. Taking this a step further, Retrv-R1 (Zhu et al., 2025b) employs reinforcement learning to optimize a step-by-step reasoning process, framing retrieval as a reasoning-driven decision-making task. Similarly, TTE-v2 (Cui et al., 2025a) extends this paradigm with a cascaded retrieval-reranking framework, using joint query-candidate reasoning to refine the initial results and leveraging the reranker as a teacher for hard negative mining. Embed-RL (Jiang et al., 2026) proposes an Embedder-Guided Reinforcement Learning (EG-RL) framework to optimize a reasoner to produce evidential traceability CoTs that better align with the embedding task. MMEmb-R1 (Wang et al., 2026) introduces pair-aware reasoning selection to identify beneficial rationales and uses reinforcement learning to adaptively invoke reasoning only when necessary. In a different direction, PLUME (He et al., 2026) internalizes reasoning into a compact latent rollout of continuous hidden states, replacing expensive explicit CoT generation while preserving the benefits of multi-step computation for efficiency.

### 3.1.4 Technical Innovations

Recent advancements in VDR embedding models can be broadly categorized as follows:

❶ **Model-level:** Innovations in this area focus on designing *novel architectures and interaction mechanisms.* A pioneering example is ColPali (Faysse et al., 2024), which first adapts the late-interaction mechanism to operate directly on document page *images*, enabling precise alignment between query and visual patches without a brittle OCR pipeline. Unveil (Sun et al., 2025a) introduces a hybrid visual-textual teacher model and then uses knowledge distillation to transfer its comprehensive understanding to an efficient, OCR-free visual-only student model.

❷ **Data-level:** This includes not only creating *large-scale, high-quality training datasets* but also developing *sophisticated data synthesis and hard negative mining strategies.* VLM2Vec (Jiang et al., 2024c) introduces MMEB benchmark, which unifies multimodal tasks into a universal ranking format. In terms of negative mining strategies, UniME-V2 (Gu et al., 2025b) proposes an "MLLM-as-a-Judge" mechanism that leverages MLLMs to assess retrieved candidates and generate soft semantic matching scores.

❸ **Training-level:** Advancements here involve exploring *novel training objectives* and *flexible paradigms beyond standard contrastive loss.* jina-embedding-v4 (Günther et al., 2025) implements a unified multi-task learning framework that simultaneously trains a model to produce both single-vector and multi-vector embeddings, while using LoRA adapters to optimize performance for different retrieval scenarios. MM-Embed (Lin et al., 2024a) introduces a modality-aware training strategy, which explicitly samples negatives that are semantically similar but have the incorrect modality, complemented by a continuous fine-tuning schedule to balance multimodal and text-only retrieval.

| Reranker | Publication | | Model | | | | Benchmark* | Resource |
|---|---|---|---|---|---|---|---|---|
| | Team | Venue | Para. | Backbone | Ranking | Multilingual | | |
| Lychee-rerank-mm (Dai et al., 2025) | HIT | ICLR'26 | 7B | Qwen2.5-VL | PO | en | MRB/MRMR | 🐙 🤗 📄 |
| UniME-V2-Reranker (Gu et al., 2025b) | MiroMind AI | AAAI'26 | 7B | Qwen2.5-VL | PA LI | en | MMEB | 🐙 🤗 📄 |
| Qwen3-VL-Reranker (Li et al., 2026a) | Alibaba | arxiv'26 | 2B/8B | Qwen3-VL | PO | >30 | MMEB-v2/JinaVDR/ViDoRe-v3 | 🐙 🤗 📄 |
| llama-nemotron-rerank-vl (Moreira et al., 2026) | NVIDIA | arxiv'26 | 3B | Llama 3.2 | LI | en | MIRACL-Vision/ViDoRe-v1/v2/v3 | 🤗 📄 |
| Rank-Nexus (Cai, 2026) | Malaya | arxiv'26 | 2B | Qwen3-VL/InternVL3 | LI | en | MMDocIR | 🐙 🤗 📄 |
| LamRA-Rank (Liu et al., 2025e) | SJTU & Xiaohongshu | CVPR'25 | 7B | Qwen2.5-VL | PO LI | en | M-BEIR | 🐙 🤗 📄 |
| DocReRank (Wasserman et al., 2025a) | WIS | EMNLP25 | 2B | Qwen2-VL | PO | en | ViDoRe-v2/Real-MM-RAG | 🐙 📄 |
| RagVL (Chen et al., 2024) | IDEA | EMNLP Finding'25 | 1B/2B/4B/13B | Qwen-VL/InternVL/LLaVA-v1.5 | PO | en | - | 🐙 📄 |
| MM-R5 (Xu et al., 2025b) | DP Tech | arxiv'25 | 7B | Qwen2.5-VL | LI | en | MMDocIR | 🐙 🤗 📄 |
| jina-reranker-m0 (Jina AI, 2025) | Jina AI | blog'25 | 2.4B | Qwen2-VL | PO | 29 | ViDoRe-v1/M-BEIR | 🤗 📄 |
| MonoQwen2-VL-v0.1 (Chaffin & Lac, 2024) | LightOn | blog'24 | 2B | Qwen2-VL | PO | en | ViDoRe-v1 | 🤗 📄 |

**Table 4:** Comparison of multimodal (document) reranker models. * indicates that only VDR-related benchmarks evaluated. In *Ranking* column, we denote POintwise/PAirwise/LIstwise design as PO / PA / LI. In *Resource* column, we denote the corresponding github codebase/huggingface/paper(*e.g.,* arxiv paper, technical report or blog) as 🐙 / 🤗 / 📄.

❹ **Efficiency-level:** This line of work challenges the "bigger is better" assumption by developing *smaller yet powerful models* and *more efficient training methods*. ModernVBERT (Teiletche et al., 2025) demonstrates a compact (250M) model based on a bidirectional encoder architecture can outperform much larger (*e.g.,* >3B) decoder-based VLMs. On training efficiency front, B3 (Thirukovalluru et al., 2025) introduces a smart batch mining strategy that pre-processes the entire dataset using graph-based detection to construct batches rich in mutual hard negatives.

## 3.2 Reranker Models

### 3.2.1 Formulation

A reranker model, denoted as $R(\cdot, \cdot)$, operates as a cross-encoder. It takes a query $q$ and a single candidate document $d_c$ from the initial retrieval stage as a combined input. By jointly processing the query's token sequence $Q = \{t_1^q, \ldots, t_M^q\}$ and the document's patch sequence $D_c = \{p_1^{d_c}, \ldots, p_N^{d_c}\}$, the model can leverage deep cross-attention mechanisms to capture fine-grained inter-modal dependencies. The output is a single scalar relevance score, $s_{\text{rerank}}$, which is used to re-sort the candidate list: $s_{\text{rerank}} = R(q, d_c) = R(\{t_1^q, \ldots, t_M^q\}, \{p_1^{d_c}, \ldots, p_N^{d_c}\})$. This deep interaction allows the model to make more accurate relevance judgments than the dot-product-based similarity used in bi-encoder.

### 3.2.2 Technical Trends

**Model Choices.** Similar to embedding models, the trend in rerankers is towards larger, MLLM-based architectures. Models such as UniME-V2-Reranker (Gu et al., 2025b) and LamRA-Rank (Liu et al., 2025e) utilize powerful Qwen2.5-VL with parameter counts reaching 7 billion. The strong multimodal understanding capabilities of MLLMs make them suited for reranking.

**Multilingual Support.** In stark contrast to the rapid adoption of multilingualism in VDR embedding models, this capability remains largely underdeveloped in the reranker space. The vast majority of existing multimodal rerankers, including DocReRank (Wasserman et al., 2025a) and MonoQwen2-VL-v0.1 (Chaffin & Lac, 2024), are English-only. The only exception is jina-reranker-m0 (Jina AI, 2025), which supports 29 languages, highlighting a significant gap for future research in developing multilingual rerankers.

### 3.2.3 Reranking Paradigms

Reranker models are typically trained using one or a combination of three main paradigms:

❶ **Pointwise:** This approach evaluates each query-document pair $(q, d_i)$ independently, predicting an absolute relevance score. The model is trained using a loss function like Binary Cross-Entropy (BCE) on the predicted score $s_i = R(q, d_i)$ against a ground-truth label $y_i \in \{0, 1\}$. It can be formulated as $\mathcal{L}_{\text{pointwise}} = \sum_i \text{BCE}(s_i, y_i)$.

❷ **Pairwise:** This method learns relative relevance by comparing pairs. Given a query $q$ and a pair of documents $(d_i, d_j)$ where $d_i$ is more relevant than $d_j$, the model is trained with a margin-based loss to ensure $R(q, d_i) > R(q, d_j)$, formulated as $\mathcal{L}_{\text{pairwise}} = \sum_{y_i > y_j} \max(0, m - (s_i - s_j))$.

❸ **Listwise:** This paradigm considers the entire list of candidate documents for a query simultaneously. It optimizes a loss function, such as ListNet's softmax cross-entropy, that directly corresponds to ranking metrics like nDCG, learning to predict the optimal ordering of the full list, formulated as $\mathcal{L}_{\text{listwise}} = -\sum_i \mathrm{P}(d_i) \log(\hat{\mathrm{P}}(d_i))$, where P and $\hat{\mathrm{P}}$ are ground-truth and predicted probability distributions over the list.

Recent rerankers have shown that combining multiple loss functions often yields superior performance. LamRA-Rank (Liu et al., 2025e) and UniME-V2-Reranker (Gu et al., 2025b) adopt hybrid training strategies that combine listwise optimization, where the model predicts the correct item's position, with pointwise or pairwise objectives that classify the relevance of individual or paired candidates. Other approaches refine a single paradigm to great effect; models like Lychee-rerank-mm (Dai et al., 2025) and DocReRank (Wasserman et al., 2025a) utilize supervised fine-tuning to frame reranking as a pointwise classification task, predicting "yes" or "no" to align with the generative nature of MLLMs. Besides, MM-R5 (Xu et al., 2025b) advances the listwise paradigm by incorporating CoT reasoning and leveraging reinforcement learning with a task-specific reward to optimize the ranked output.

### 3.3 RAG Pipeline & Agentic System

#### 3.3.1 Formulation

The integration of a VDR embedding model $E(\cdot)$ and a reranker $R(\cdot, \cdot)$ into a RAG pipeline can be formulated as a multi-stage process. Given a query $q$ and a corpus $C$, the process unfolds as follows:

❶ **First-Stage Retrieval:** An embedding model $E$ is used to efficiently retrieve an initial set of $k$ candidate documents $C_k$ from the corpus $C$ based on embedding similarity. It can formulated as: $C_k = \underset{d \in C}{\text{Top-k}} \left( s(E(q), E(d)) \right).$

❷ **Second-Stage Reranking:** A reranker model $R$ then refines this candidate set by computing a more accurate relevance score for each document, producing a final ranked list $C_j'$ (where $j \leq k$), formulated as $C_j' = \underset{d \in C_k}{\text{Top-j}} \left( R(q, d) \right).$

❸ **Augmented Generation:** Finally, a generative MLLM, $G(\cdot)$, synthesizes an answer $a$ by conditioning on both the original query $q$ and the context provided by the retrieved and reranked documents $C_j'$, formulated as $a = G(q, C_j')$.

In an **Agentic system**, this process becomes dynamic and iterative. An agent $A$ uses the VDR model as a tool. At each step $t$, based on the query $q$ and internal state (or history) $h_t$, the agent generates an action $act_t = A(q, h_t)$. If the action is a retrieval query $q_t$, the VDR system is invoked to retrieve evidence $C_t = \text{Retrieve}(q_t)$. The agent then updates its state $h_{t+1} = h_t \cup \{q_t, C_t\}$ and decides on the next action.

#### 3.3.2 Current Paradigms and Key Trends

The integration of VDR into RAG and Agentic systems has moved beyond simple document fetching, evolving into sophisticated frameworks that emulate human-like reasoning and interaction (Li et al., 2025h; Arya & Gaud, 2025). This evolution is characterized by several key trends, progressing from foundational end-to-end pipelines to complex, iterative, and deeply aware reasoning workflows.

❶ **Foundational Multimodal RAG Pipelines.** The most fundamental shift has been the move from brittle OCR-based textual RAG to robust, end-to-end multimodal pipelines that process documents as visual inputs. This paradigm preserves critical layout and graphical information often lost in text extraction. A prime example is M3DocRAG (Cho et al., 2024), which establishes a flexible framework for multi-page and multi-document question answering by directly retrieving relevant page images for a multimodal generator, forming a foundational approach for subsequent innovations.

❷ **Expansion of Interaction Modalities.** Building on the visual-centric pipeline, researchers are expanding the interaction modalities beyond traditional text-based queries to create more natural and accessible

interfaces. A pioneering work in this direction is TextlessRAG (Xie et al., 2025), which introduces a fully "textless" pipeline that directly processes speech queries and generates spoken answers without any explicit Automatic Speech Recognition (ASR) or Text-to-Speech (TTS) steps, showcasing the potential to significantly broaden the application scenarios of VDR.

❸ **Emergence of Agentic and Iterative Reasoning Workflows.** A dominant trend is the replacement of static, linear RAG pipelines with dynamic, agentic systems that perform multi-step, iterative reasoning. These systems decompose complex queries and progressively refine evidence, mimicking human research processes.

- **Task Decomposition and Collaboration:** Many frameworks now employ a "society of agents," where specialized agents collaborate to solve a problem. For instance, MDocAgent (Han et al., 2025b) utilizes a team of five agents (e.g., General, Critical, Text, and Image agents) to synthesize insights from different modalities. Similarly, ViDoRAG (Wang et al., 2025b) introduces a coarse-to-fine workflow where a "Seeker" agent hunts for relevant images and an "Inspector" agent provides detailed feedback, enabling iterative evidence refinement.

- **Iterative Deep Research Workflows:** This agentic concept is scaled further in systems designed for "deep research." Doc-Researcher (Dong et al., 2025c) implements a comprehensive multi-agent framework with a "Planner" for query decomposition and a "Searcher-Refiner" loop that iteratively gathers and filters evidence across multiple documents and granularities (*e.g.,* chunks, pages, or summaries).

❹ **Enhancing Core RAG Components with Advanced Mechanisms.** Beyond structuring the workflow, significant innovation is occurring within the core retrieval and generation components themselves to make them more intelligent and aware.

- **Holistic Knowledge Retrieval and Fusion:** To address the challenge that standard retrieval often misses nuanced information, new methods are designed for more comprehensive knowledge extraction. HKRAG (Tong et al., 2025) explicitly models and retrieves both "salient" and "fine-print" knowledge using a hybrid masking retriever and an uncertainty-guided generator. In a similar vein, VisDoMRAG (Suri et al., 2025) runs parallel textual and visual RAG pipelines and then employs a consistency-constrained fusion mechanism to intelligently integrate their outputs.

- **Active Visual Perception and Learning:** The most advanced systems empower agents with the ability to actively interact with retrieved visual content. VRAG-RL (Wang et al., 2025c) pioneers this by defining a visual perception action space that allows an agent to perform actions like "crop" and "zoom" on retrieved images. This interactive process is optimized using reinforcement learning, enabling the agent to actively seek out fine-grained details in a coarse-to-fine manner, much like a human analyst.

In summary, the application of VDR in RAG and Agentic systems is rapidly maturing from a simple retrieval-and-generation process to highly dynamic, interactive, and collaborative workflows. The field is pushing towards systems that not only find relevant documents but also intelligently reason, synthesize, and interact with multimodal information to solve complex problems (Ashraf et al., 2025; Zhang et al., 2025i; Yan et al., 2025c; Su et al., 2025a).

### 3.3.3 The Evolutionary Synergy of VDU and VDR

The boundary between Visual Document Understanding (VDU) and VDR is rapidly dissolving in the MLLM era. Modern RAG pipelines are evolving into autonomous agentic systems capable of perception, strategic planning, and iterative refinement.

**VDU Trends in RAG and Agentic Systems.** Recent breakthroughs emphasize moving from single-turn OCR-based retrieval to multi-step reasoning-driven navigation. (Sourati et al., 2025) introduces LAD-RAG, which leverages an LLM agent to dynamically interact with a symbolic document graph and neural indices, capturing structural dependencies missed by dense embeddings. (Yu et al., 2025b) proposes MACT, a

collaborative framework that decomposes document intelligence into planning, execution, and judgment agents, implementing a self-correction loop via procedural scaling. To address evidence sparsity, MHier-RAG (Gong et al., 2025) facilitates multi-granularity reasoning by retrieving parent pages and document summaries, while MoLoRAG (Wu et al., 2025) constructs page graphs to navigate logical connections beyond surface-level semantic similarity. For fine-grained localization, DocLens (Zhu et al., 2025a) employs a tool-augmented "zoom-in" strategy where agents locate specific visual elements like tables or charts within retrieved pages. Finally, SLEUTH (Liu et al., 2025b) optimizes the input quality through context engineering, utilizing page-screening agents to filter visual noise and construct evidence-dense multimodal contexts.

**Comparison between VDR and VDU.**  Table 5 illustrates the multi-dimensional differences and the emerging convergence between retrieval and understanding tasks. While VDR focuses on efficient large-scale candidate localization, VDU emphasizes deep reasoning. The integration of the two, as seen in recent agentic frameworks, enables systems to perform "Retrieval as Understanding."

| Dimension | Visual Document Retrieval (VDR) | Visual Document Understanding (VDU) | Convergence Trend |
|---|---|---|---|
| **Core Objective** | Candidate page localization | Information extraction & reasoning | R → U Integrated |
| **Granularity** | Coarse (Document/Page-level) | Fine (Element/Token-level) | Hierarchical Indexing |
| **Computation** | High (Scanning entire corpus) | Low (Deep processing of Top-K) | Adaptive Resource Scaling |
| **Key Technique** | Late Interaction, Multi-vector Indexing | Visual CoT, Multimodal Fusion | Agent-driven Seek-then-Verify |
| **Main Metric** | nDCG, Recall@K | Accuracy, F1-score | Source Attribution Accuracy |

**Table 5:** Multi-dimensional comparisons between VDR and VDU. We highlight that recent Agentic systems are bridging the gap through iterative reasoning and dynamic filtering.

**Reflections on VDR: Toward Cognitive Discovery.**  Future VDR systems should evolve from passive semantic matchers to active cognitive navigators. The integration of reinforcement learning (as in VRAG-RL (Wang et al., 2025c)) and symbolic-neural fusion (as in LAD-RAG (Sourati et al., 2025)) suggests that the next frontier of multimodal document intelligence lies in the ability to understand the "logical topology" of a document. Rather than just returning snippets, future retrievers will actively discover implicit contradictions and synthesize cross-document insights, transforming VDR into a tool for complex decision support in expert domains.

## 4 Challenges & Outlook

As shown in Figure 4, A truly *effective*, *efficient*, and *interactive* system is fraught with persistent challenges. The ultimate goal extends beyond simple document matching to enabling nuanced, real-time document intelligence that is scalable and trustworthy. Achieving this vision requires addressing critical bottlenecks across the entire VDR ecosystem. These challenges span from the foundational level of data (▷ Section 4.1) and model architectures (▷ Section 4.2) to the practical imperatives of performance efficiency (▷ Section 4.3) and system interactivity (▷ Section 4.4), and culminate in the higher-order needs for understanding model scaling behavior (▷ Section 4.5). The following sections delineate five core challenges and outline promising future directions to navigate these frontiers.

### 4.1 Expanding the Data Frontier

The foundation of robust VDR systems is high-quality, diverse, and challenging data, yet the current landscape of benchmarks and training sets presents a significant bottleneck to progress. A primary limitation is the lack of diversity in language, domain, and document structure. While recent benchmarks like Jina-VDR (Günther et al., 2025) and MIRACL-VISION (Osmulski et al., 2025) have made commendable strides in multilingual support, the majority of available resources remain predominantly English-centric and confined to general-domain web documents. Furthermore, most benchmarks focus on retrieving single, self-contained pages, failing to capture the complex, multi-hop, and cross-document reasoning required for genuine information synthesis—a gap that pioneering benchmarks like MR2-Bench (Zhou et al., 2025a) and MRMR (Zhang et al., 2025e) are only beginning to explore. A more insidious issue lies in data provenance; the heavy reliance on VLM-generated queries for training data, a common practice in datasets, risks creating a hermetic feedback

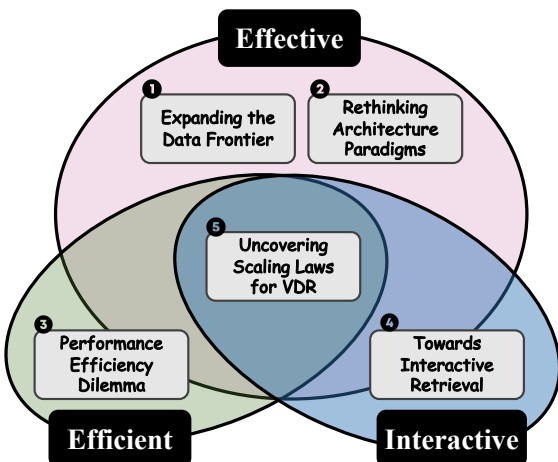

**Figure 4:** Big picture of future challenges in VDR domain.

loop where models are evaluated on the same kind of logic they were trained on, inflating performance metrics without guaranteeing real-world generalization. This is compounded by the potential for data leakage between large, web-crawled training sets and evaluation benchmarks, which casts a shadow over the true robustness of current models (Apicella et al., 2025; Lilja et al., 2024).

Addressing these data frontiers requires a multi-pronged, forward-looking strategy that moves beyond mere data scaling. A primary imperative is the creation of large-scale benchmarks that are ***not only multilingual but also multi-domain***, extending into specialized corpora like legal contracts, financial reports, and medical records, which demand expert-level nuance (Tang & Yang, 2024; Cao, 2024; Chung et al., 2025). To cultivate ***reasoning-intensive capabilities***, future data curation can leverage agentic frameworks to automatically generate complex, multi-hop query-evidence chains that span multiple documents, simulating the realistic research workflows envisioned by systems like Doc-Researcher (Dong et al., 2025c). To break the cycle of synthetic data bias and enhance authenticity, a pivot towards ***incorporating real human queries*** is essential; this could be achieved by mining anonymized search logs or utilizing human-in-the-loop annotation platforms to capture the ambiguity and complexity of genuine user intent. Finally, to ensure ***robust and fair evaluation***, the community must adopt stricter data governance practices. This includes establishing "firewalled" test sets derived from entirely held-out web crawls or proprietary document collections and developing standardized protocols for data contamination detection, thereby fostering a more reliable assessment of model generalization and pushing the field towards true document intelligence (Song et al., 2024; Xu et al., 2024; Samuel et al., 2025).

## 4.2 Rethinking Architectural Paradigms

The multimodal retrieval field has largely converged on a paradigm centered around contrastively-trained discriminative models, typically bi-encoder architectures that learn to maximize the similarity between positive query-document pairs while minimizing it for negative ones. While highly effective for semantic matching, this approach has inherent limitations. It primarily treats powerful MLLMs as static feature extractors (Mischler et al., 2024; Jeong et al., 2024), underutilizing their sophisticated generative and reasoning capabilities, which can lead to information bottlenecks where fine-grained details or implicit document semantics are lost during the compression into a fixed-size embedding. Moreover, this "one-size-fits-all" strategy forces a single, dense model to act as a generalist for an incredibly diverse array of document structures—from text-heavy reports and dense tables to complex charts and forms—limiting both performance specialization and computational efficiency (Gan et al., 2025b; Fan et al., 2024; Lo et al., 2025).

To overcome these limitations, future research is pivoting towards two promising architectural shifts: autoregressive retrieval and the integration of Mixture of Experts (MoE). The first paradigm ***reframes retrieval from a purely discriminative task to a generative one*** (Deng et al., 2025a; Xiong et al., 2024). For instance, future models could adopt an "embed-via-answering" framework, where embeddings are derived from

generating a concise answer to an instruction, as pioneered by InBedder (Peng et al., 2024). More advanced paradigms may even involve generating sequences of non-human-readable "soft tokens" that iteratively refine the embedding, a concept introduced by GIRCSE (Tsai et al., 2025) which learns to "speak an embedding language" and shows improved quality with more generation steps at test time. This generative-representational synergy is also validated by frameworks like GritLM (Muennighoff et al., 2024), which successfully unifies generative and embedding tasks within a single model, demonstrating that these capabilities can coexist without performance degradation. We summarize and compare the representative works from text-only and multimodal generative retrieval in Table 6.

| Representative Works | Modality | Core Paradigm | Generated Content | Embedding Source | Training Objective |
|---|---|---|---|---|---|
| **InBedder** (Peng et al., 2024) | Text | Embed-via-Answering | Concise Answer | Hidden state of the 1st generated token | Instruction Tuning (QA) |
| **GIRCSE** (Tsai et al., 2025) | Text | Iterative Refinement | Soft Tokens (Non-readable) | Pooled hidden states of generated soft tokens | Iterative Contrastive Refinement (ICR) |
| **GritLM** (Muennighoff et al., 2024) | Text | Task Unification | Text Response or Embedding | Mean pooling over input with bidirectional attention | Joint Contrastive + LM Loss |
| **RGE** (Liu et al., 2025a) / **TTE** (Cui et al., 2025b) | Multimodal | Think-then-Embed | Reasoning Trace (CoT) | Hidden state conditioned on generated reasoning | Joint LM (on trace) + Contrastive Loss |
| **CAFe** (Yu et al., 2025a) / **VladVA** (Ouali et al., 2025) | Multimodal | Joint Training | N/A (for retrieval) | Last token's hidden state | Joint Contrastive + Autoregressive Loss |
| **Retrv-R1** (Zhu et al., 2025b) | Multimodal | Reasoning-driven Selection | Reasoning trace leading to a selection | Not an embedding model; selects best candidate | SFT + Reinforcement Learning (GRPO) |

**Table 6:** A multi-dimensional comparison of representative works in **generative retrieval**. These approaches move beyond static encoding by leveraging the generative capabilities of (M)LLMs. They differ in their core paradigm, the nature of the generated content (from concise answers to reasoning traces), and their training objectives, which range from joint contrastive-generative losses to reinforcement learning.

Concurrently, ***the MoE architecture offers a path to efficient specialization***. For instance, different experts can be trained to handle distinct document modalities (*e.g.,* text, tables, audio), a direction explored by Uni-MoE (Li et al., 2025j;i) and M3-JEPA (Lei et al., 2024), which use multi-gate MoE to disentangle modality-specific and shared information. This architecture also unlocks novel embedding sources, as demonstrated by MOEE (Li & Zhou, 2024), which shows that the expert routing weights themselves can serve as a potent, complementary embedding signal. Together, these architectural innovations promise to evolve VDR from static matching towards more dynamic, specialized, and generative systems.

## 4.3 Performance-Efficiency Dilemma

The high accuracy of state-of-the-art multi-vector VDR models stems directly from their fine-grained representation paradigm, where each document page is encoded into hundreds or even thousands of patch-level embeddings (Faysse et al., 2024). While this enables precise query-to-patch matching, it creates a significant performance-efficiency dilemma. The primary issue is the prohibitive storage overhead; for instance, a medium-sized document of just 50 pages can require approximately 10 MB of storage for its embeddings alone (Ma et al., 2025), rendering the large-scale deployment of such models economically and practically challenging. Furthermore, this abundance of vectors increases online computational costs during the late-interaction scoring stage. This dilemma has spurred research into embedding reduction techniques to create more compact yet effective document representations.

Future solutions to this dilemma are evolving from post-hoc compression to built-in adaptability. The two primary post-hoc strategies are ***embedding pruning and merging*** (Yan et al., 2026b; Cha et al., 2026; Qin et al., 2026; Pony et al., 2026; Liu et al., 2026a). Pruning aims to discard redundant embeddings. While early heuristic-based methods proved unstable (Liu et al., 2024; Lassance et al., 2021; 2022), more sophisticated approaches like DocPruner (Yan et al., 2025d) use attention scores to adaptively prune 50-60% of patches per document with minimal performance loss. Alternatively, merging or clustering aggregates similar patches, a method that some argue is more suitable as it retains information rather than discarding it. For example, Light-ColPali (Ma et al., 2025) employs hierarchical clustering to merge embeddings, while HEAVEN (Kim et al., 2025a) uses visually-summarized pages to create a compact first-stage index.

A more forward-looking paradigm is ***Matryoshka Representation Learning (MRL)*** (Kusupati et al., 2022), which trains embeddings to be inherently flexible. MRL ensures that a single high-dimensional embedding contains a hierarchy of smaller, nested embeddings that are also effective. The flexible Matryoshka embedding settings have applied in top performing text embedding models (*e.g.,* Qwen3 Embedding (Zhang et al., 2025h), Gemini Embedding (Lee et al., 2025b), KaLM-Embedding-V2 (Zhao et al., 2025), EmbeddingGemma (Vera et al., 2025), jina-embeddings-v3 (Sturua et al., 2024), jina-colbert-v2 (Jha et al., 2024)) across diverse real-world tasks (Lai et al., 2024; Wang et al., 2024c; Nacar & Koubaa, 2025; Hanley

& Durumeric, 2025; Fu et al., 2025). Formally, given an encoder $\mathcal{E}$, a document page $d$ is mapped to a full embedding $\mathbf{e}_d = \mathcal{E}(d) \in \mathbb{R}^D$. MRL trains the model on a set of nested dimensions $M = \{m_1, m_2, \ldots, m_k\}$ where $m_1 < m_2 < \ldots < m_k = D$, such that each prefix $\mathbf{e}_d^{[m_i]} \in \mathbb{R}^{m_i}$ (the first $m_i$ components of $\mathbf{e}_d$) is a valid representation. The training objective $\mathcal{L}_{\text{MRL}}$ sums a standard representation loss $\ell(\cdot, \cdot)$ over all nested dimensions:

$$\mathcal{L}_{\text{MRL}}(q, d^+) = \sum_{m_i \in M} w_i \cdot \ell \left( \mathcal{E}(q)^{[m_i]}, \mathcal{E}(d^+)^{[m_i]} \right) \tag{11}$$

where $q$ is a query, $d^+$ is a positive document, and $w_i$ are loss weights. This allows practitioners to truncate stored embeddings at inference time, providing a flexible dial to balance performance and efficiency. Building on this, 2DMSE (Li et al., 2024c) extends this elasticity to model depth, while frameworks like Starbucks (Zhuang et al., 2024) and MatTA (Verma et al., 2025) refine the training process to bridge the performance gap with individually trained models. The Matryoshka principle has also been generalized to new granularities and architectures: M3 (Cai et al., 2024) and MQT (Hu et al., 2024) apply it to the number of visual tokens in LVLMs; Matryoshka Re-Ranker (Liu et al., 2025g) enables flexible depth and sequence length for re-ranking; and MatMamba (Shukla et al., 2024) integrates it into State Space Models. The application of MRL has further expanded to diverse tasks such as improving feature hierarchy in Sparse Autoencoders (Bussmann et al., 2025) and enabling multi-precision models via Matryoshka Quantization (Nair et al., 2025). Finally, innovations in the training paradigm itself, such as the sequential learning in SMEC (Zhang et al., 2025a) and the post-hoc tuning of Matryoshka-Adaptor (Yoon et al., 2024), are making the creation of these flexible embeddings more efficient and accessible. Besides, recent works also focus on sparse coding (Wen et al., 2025; Guo et al., 2026) and isolation kernel (Zhang et al., 2026b) for more lightweight representation learning. We compare the representative works above in Table 7.

**Table 7:** A multi-dimensional comparison of representative works in **Matryoshka Representation Learning (MRL)**. The principle of nested, adaptable representations has been extended from embedding dimensions to model depth, token counts, and even quantization bit-widths, addressing a wide range of tasks from efficient retrieval to model interpretability.

| Representative Works | Modality | Target Task | Matryoshka On | Core Innovation |
|---|---|---|---|---|
| **MRL** (Kusupati et al., 2022) | Multimodal | Classification / Retrieval | Embedding Dimension | Foundational concept of nested embeddings. |
| **2DMSE** (Li et al., 2024c) | Text | STS / Retrieval | Embedding Dimension & Model Depth | Extends MRL to two dimensions for greater flexibility. |
| **Starbucks** (Zhuang et al., 2024) | Text | STS / Retrieval | Embedding Dimension & Model Depth | Improves 2DMSE to match individually trained models. |
| **MatTA** (Verma et al., 2025) | Text | General Tasks | Model Depth & Width (FFN) | Teacher-TA-Student distillation for elastic student models. |
| **M3** (Cai et al., 2024) / **MQT** (Hu et al., 2024) | Multimodal | VQA / Reasoning | Number of Visual Tokens | Enables adaptive visual granularity in LVLMs. |
| **Matryoshka Re-Ranker** (Liu et al., 2025g) | Text | Re-ranking | Model Depth & Sequence Length | Creates flexible re-rankers with configurable architecture. |
| **SMEC** (Zhang et al., 2025a) | Multimodal | Retrieval / Classification | Embedding Dimension | Sequential training to mitigate gradient variance. |
| **Matryoshka-Adaptor** (Yoon et al., 2024) | Multimodal | Retrieval | Embedding Dimension | Post-hoc tuning to impart Matryoshka properties. |
| **MatMamba** (Shukla et al., 2024) | Multimodal | LM / Classification | Hidden Dimension & Heads | Generalizes MRL to State Space Model (SSM) architectures. |
| **Matryoshka SAE** (Bussmann et al., 2025) | Model Activations | Interpretability | SAE Dictionary Size | Mitigates feature splitting in Sparse Autoencoders. |
| **Matryoshka Quantization** (Nair et al., 2025) | Model Weights | Model Compression | Quantization Bit-width | Leverages nested structure of integer data types. |

## 4.4 Towards Interactive Retrieval

Integrating VDR with agentic systems holds immense potential, particularly for complex scenarios like *Deep Research*, where a query requires iterative evidence gathering from a vast candidate pool (Java et al., 2025; Chen et al., 2025e; Huang et al., 2025b; Zhang et al., 2025f). The current challenge, however, is that most agentic systems treat VDR models as passive, reactive tools rather than active, strategic partners. These agents typically operate within a predefined, reactive loop of reasoning and acting, which limits their ability to dynamically formulate or adapt a high-level retrieval strategy. For instance, while frameworks like WebThinker (Li et al., 2025g) and DeepResearcher (Zheng et al., 2025b) enable agents to interleave thought with web search actions, they often function as sophisticated *tool-executors* that reactively process information rather than as *research strategists* that proactively plan a multi-step information-seeking campaign. This limitation becomes particularly evident in complex research tasks, such as generating structured reports with dynamic outlines as demonstrated by WebWeaver (Li et al., 2025k) or creating text-chart interleaved content as in Multimodal DeepResearcher (Yang et al., 2025b), where the retrieval process itself must be strategically guided to gather diverse and structured evidence.

Future work must focus on the ***co-design of agents and VDR tools to foster a more organic and strategic synergy***. A primary direction is to empower agents with explicit retrieval planning capabilities, enabling them to decompose a high-level query into a multi-step, multi-granularity retrieval plan. This

moves beyond simple tool use towards strategic orchestration. For example, a planner agent, inspired by the hierarchical structure of Cognitive Kernel-Pro (Fang et al., 2025), could adaptively select *retrieval granularity*—choosing between document summaries, full pages, or specific visual elements—and delegate these sub-tasks to specialized agents. To handle the inevitable "dead ends" in complex research, agents must also develop self-correction mechanisms. Frameworks like DeepAgent (Li et al., 2025f), with its autonomous memory folding, offer a powerful blueprint by allowing an agent to "take a breath," discard a failed exploration path, and restart from a consolidated memory state. To effectively learn such complex strategies, agents require more fine-grained feedback. This can be achieved through mechanisms like the Atomic Thought Rewards (ATR) proposed in Atom-Searcher (Deng et al., 2025b), which uses a reasoning reward model to provide process-level supervision on the agent's retrieval strategy. Finally, for massive-scale research, parallel exploration frameworks like the Research-Synthesis approach in WebResearcher (Qiao et al., 2025) and Tongyi DeepResearch (Team et al., 2025b), where multiple agents conduct concurrent searches, can transform interactive retrieval into a scalable, collaborative process. The ultimate goal is to evolve the VDR system from a passive tool into an active, *strategic partner* in the knowledge discovery process.

### 4.5 Uncovering Scaling Laws for VDR

While scaling laws (predictable power-law relationships between model performance, size, and data volume) are well-documented for general-purpose LLMs (Alabdulmohsin et al., 2022; Zhang et al., 2024a; Xiao et al., 2025a; Pearce et al., 2024), their application to the specialized domain of VDR remains a largely unexplored and complex frontier. The core challenge is that the scaling behavior in VDR is not a straightforward extrapolation of model size or data quantity. On the model axis, naively increasing parameter count does not guarantee proportional performance gains. Larger models, without proper fine-tuning, can exhibit greater embedding anisotropy, which paradoxically harms retrieval performance in zero-shot settings by compressing the effective embedding space, even as they capture richer features for transfer tasks (Jiang et al., 2024a; Wu et al., 2026). This complexity is mirrored on the data axis, where simply increasing the volume of raw documents is insufficient. The effectiveness of VDR models is deeply tied to the quality and diversity of the training pairs. Recent studies powerfully illustrate this, showing that models trained on smaller, high-quality synthetic datasets can significantly outperform those trained on orders-of-magnitude more, but less curated, data. For example, both mmE5 (Chen et al., 2025b) and MegaPairs (Zhou et al., 2025b) demonstrated that superior performance can be achieved with a fraction of the data if it is diverse and of high quality, underscoring that scaling in VDR is a nuanced interplay between model capacity and data sophistication, rather than a simple numbers game.

To systematically navigate this challenge, future research must pivot from ad-hoc scaling to **establishing formal, VDR-specific scaling laws, adapting methodologies from the dense retrieval domain** (Fang et al., 2024). A foundational step is to adopt continuous and sensitive evaluation metrics, such as contrastive entropy, which can more accurately capture subtle performance changes compared to discrete ranking metrics like nDCG, thus enabling precise power-law curve fitting for model size ($N$) and data volume ($D$). Concurrently, advancing the "data" axis of the scaling law requires moving beyond simple data augmentation to sophisticated, document-aware synthetic data generation. This involves leveraging MLLMs not just for query generation but for creating entire ecosystems of diverse training tasks (Wang et al., 2024a), mining heterogeneous relationships between documents using multiple similarity models (*e.g.,* visual-semantic and visual-pattern correlations) (Zhou et al., 2025b), and implementing rigorous quality control through mechanisms like single-pass multi-aspect interpretation and self-refinement (Chen et al., 2025b). By combining a formal scaling law framework with advanced data synthesis, the VDR community can quantitatively model the trade-offs between investing in larger models versus more extensive data annotation. This will enable more efficient resource allocation and pave the way for building maximally effective VDR systems under practical budget constraints.

## 5 Conclusion

This survey systematically discusses VDR landscape, categorizing the evolution of benchmarks and methodologies (embedding, reranker, and integration with RAG and Agents). Current triumphs push the boundaries

of fine-grained matching, but the field's trajectory is shifting towards more complex document intelligence. Navigating the future frontiers of data complexity, architectural innovation, efficiency, and interactivity will be critical for realizing the full potential of VDR as a cornerstone of multimodal document intelligence.

## 6 Broader Impact and Societal Considerations

While this survey focuses on the technical advancements of VDR in the MLLM era, the deployment of such systems carries significant broader societal implications. As VDR transitions from academic benchmarks to real-world deployment in legal, medical, and enterprise workflows, we must critically evaluate its impact on accountability, equity, and environmental sustainability.

**Transparency, Accountability, and Trust.** A primary concern in deploying VDR systems, particularly in high-stakes domains, is the "black-box" nature of multimodal reasoning. When retrieval decisions are based on opaque, latent-space alignments, the inability to provide clear, human-interpretable rationales (or citations to evidence) complicates auditability. To ensure responsible deployment, the development of VDR should not only prioritize retrieval accuracy but also emphasize traceable reasoning. Systems that integrate "retrieval-as-reasoning", where the model provides an explicit chain-of-thought or highlights the specific visual/textual evidence, are essential for establishing trust and accountability, especially in regulatory contexts where document decisions must be legally defensible.

**Mitigating Bias and Promoting Multilingual Equity.** The VDR landscape remains heavily skewed towards English-centric datasets and models. If left unchecked, this "linguistic and cultural bottleneck" threatens to marginalize non-English speaking communities and industries, potentially reinforcing existing global inequities in information access. The surge in multilingual benchmark development is a necessary step, but technical efforts must also address inherent biases within foundation models. Retrieval systems that unknowingly favor dominant cultural or Western-centric document formats risk systematic information omission. We advocate for research that explicitly models fairness across linguistic and structural diversities to ensure that VDR is a globally inclusive technology.

**Environmental Impact of Large-Scale Intelligence.** The transition from traditional, lightweight OCR-based search to massive-scale VDR models based on multi-vector late-interaction paradigms introduces a significant performance-efficiency dilemma. While these models offer unprecedented semantic depth, they impose substantial computational costs in terms of memory storage, index maintenance, and GPU-intensive inference. In an era where AI-related energy consumption is under intense scrutiny, the pursuit of performance must be balanced with environmental responsibility. Developing sparse, efficient, and Matryoshka-style representations is not merely a technical optimization, it is a critical requirement to ensure that the widespread adoption of multimodal intelligence does not come at the expense of an unsustainable carbon footprint.

**Societal Safeguards and Data Integrity.** The ability of VDR models to extract fine-grained, structured information from unstructured PDFs, including sensitive personal data, private business intelligence, and restricted intellectual property, poses risks of unintended information disclosure. The shift toward agentic RAG systems that "browse" and synthesize evidence across vast document stores exacerbates the danger of bypassing existing security controls. Therefore, we underscore the necessity of robust data governance, including fine-grained access control and privacy-preserving retrieval architectures, as foundational requirements for future VDR systems.

In summary, the evolution of VDR toward more sophisticated, autonomous, and reasoning-intensive workflows necessitates a proactive approach to societal safeguards. We hope this perspective encourages the community to view technical robustneess, interpretability, and ecological sustainability not as peripheral concerns, but as integral pillars of responsible multimodal document intelligence.

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
