# OpenReview forum: "Unlocking Multimodal Document Intelligence: From Current Triumphs to Future Frontiers of Visual Document Retrieval"
_TMLR — Withdrawn by Authors_

### Review · Reviewer_u7S3 · 2026-06-06

**Summary Of Contributions:**

The authors set out to survey visual document retrieval (VDR) in the era of MLLMs. The task itself is worthwhile, since VDR has drawn considerable attention recently. The paper has three parts. It maps the evaluation landscape from a benchmark perspective (§2), organizes methods into three branches of embedding, reranker, and RAG-Agent (§3), and distills open problems along five directions (§4), with an additional Broader Impact section (§6).

I recognize the effort that went into this. For a survey, though, the value rests on three things: an accurate bibliography, a clean organization, and insightful synthesis. In my view the paper does not reach a publishable standard on any of them. Details follow.

**Audience:**

Yes

**Audience Explanation:**

**Yes, but only barely.**

VDR is an active area, so on technical grounds I answer Yes.

As a survey, however, it adds relatively little beyond what an attentive reader would already take from the primary sources. Most of it is enumeration plus roll call, a sentence or two per method along the lines of X did A and Y did B, which amounts to threading a pile of abstracts into tables.

The real value of a survey lies in synthesis, in seeing patterns others have missed, in naming the tensions in the field, and in using a taxonomy that reorganizes how we think so as to bring order to the mess. I answer Yes as the rubric requires, but that does not mean I find the paper compelling enough to draw the community in.

**Broader Impact Concerns:**

My main concern relates to the hallucinated citations noted at the start.

**Claims And Evidence:**

No

**Claims Explanation:**

**(1) [SEVERE] The reference list contains hallucinated or fabricated bibliographic information.**

Here are some of the hallucinations I found, including but not limited to the following.

- For MMEmb-R1 (arXiv:2604.06156), the fifth author is in fact **Xiao Liang**, but the paper writes **Xiaodan Liang**.
- For the flagship RAG survey by Gao et al. (arXiv:2312.10997), the author list ends with `Haofen Wang, and Haofen Wang`, so the same person appears twice, the real second to last author **Meng Wang** is dropped, and `Yi Dai` is rendered as `Yixin Dai`.
- For GritLM (Muennighoff et al., arXiv:2402.09906), the paper lists the venue as the Thirteenth International Conference on Learning Representations, **2024**, yet the thirteenth ICLR was held in **2025**.
- For Structural Anchor Pruning (arXiv:2601.20107), the first author is in fact **Zhuchenyang Liu**, but the authors write Zhu Liu.
- For Col-Bandit (arXiv:2602.02827), the second author **Adi Raz Goldfarb** is truncated to `Adi Raz`.
- For Multi-Vector Similarity Search (arXiv:2604.02815), the third author **Hengxin Zhang** is written as Heng Zhang.
- For RaDeR, the EMNLP pages are in fact `19970-19997`, but the authors write `19981-20008`.
- For MoLoRAG, the EMNLP pages are in fact `14024-14045`, but the authors write `14035-14056`.
- For SMEC, the EMNLP pages are in fact `26209-26222`, but the authors write `26220-26233`.

Whatever the cause, authors are responsible for the accuracy of the work they cite, and for a survey this is the most basic floor to clear. This kind of problem is also taken very seriously by the research community right now, and out of care for the TMLR community I consider it severe enough to warrant rejection on its own.

---

**(2) The novelty is overstated.**

The claimed first comprehensive survey is, in my view, overstated, and it comes with an underestimation of existing work.

In the Abstract and §1 the authors repeatedly describe the work as the first comprehensive survey of the VDR landscape through the lens of the MLLM era, and they support this with Table 1, which assigns checkmarks to twelve competing surveys one by one. I went through that table and found a consistent bias in how the checkmarks are assigned, always in a direction that flatters the authors.

- **A strong competitor that should appear is missing.** Abootorabi et al., *Ask in Any Modality: A Comprehensive Survey on Multimodal RAG* (ACL 2025 Findings), covers retrieval, RAG, and agents, which is exactly the axis the authors use to argue for their distinctiveness. For a survey that calls itself comprehensive to omit it suggests either an incomplete search or a selective omission, and neither is reassuring.
- **Gao et al. (2025) is underestimated.** Gao et al. (2025) is itself a retrieval oriented survey that covers embedding models, rerankers, the ColPali family, efficiency and compression, and retrieval benchmarks, only framed within a narrative that runs from multimodal RAG to document understanding. Marking it as `U` in the Scope column, and using that to set up the authors as the only `R`, is an oversimplified and somewhat misleading dichotomy. The real difference between the two papers is one of narrative emphasis, not that one does retrieval and the other does not.
- **The closest work, Zhang (2025), AACL-IJCNLP 2025, is not distinguished.** This is a VDR retrieval survey with heavily overlapping scope. A claim of being first most needs to draw a clear line against this work, yet the main text only gives it a checkmark and offers no substantive comparison that explains what the present paper adds.

Novelty rests on having read the literature fully and on showing what the new work adds beyond it. Claiming first comprehensive while at the same time omitting and underestimating prior work is not, in my view, appropriate.

---

**(3) Some descriptions of cited methods are wrong.**

For a survey, faithfully and accurately restating existing work is also basic and important, yet the authors have problems here. I give one example.

- §3.1.3 states that E5-V pioneers a training free approach that elicits embeddings from the MLLM vocabulary space, but the core of E5-V is contrastive training on single modality text pairs, not a training free method.

---

**(4) The taxonomy is not clearly organized.**

§3 sorts methods into three so called paradigms of embedding, reranker, and RAG-Agent. The problem is that this collapses two orthogonal axes, namely what a model is versus how a model is deployed, into a single flat list of three items that are all labeled as paradigms, and it never declares how many organizing axes are in play. The paper itself, in §3.3.1, shows that the third branch is downstream of the first two. A good survey taxonomy should pick a single organizing principle for each axis and state explicitly how many axes there are. As written, this scheme does not help the reader make sense of the field.

**Requested Changes:**

Let me say this up front. What follows is not a checklist that, once addressed, would make the paper acceptable. In its current state I believe it needs to be sent back for a substantial rework rather than a single round of minor or major revision. I still list the issues to help the authors improve the work.

1. Rebuild and re-verify the entire reference list.
2. Re-argue or withdraw the first comprehensive claim.
3. Restructure the taxonomy.
4. Inject genuine synthesis.

---

### Review · Reviewer_ByKH · 2026-06-09

**Summary Of Contributions:**

This paper presents a comprehensive survey of Visual Document Retrieval in the era of Multimodal Large Language Models. It distinguishes this task from traditional image retrieval by emphasizing the dense text, complex layouts, and detailed semantic dependencies of visual documents. The authors systematically trace the field by examining benchmarks, evaluation metrics, and the shift toward complex reasoning evaluations. They categorize current methodologies into multimodal embedding models, rerankers, and their integration into Retrieval Augmented Generation and Agentic systems. Finally, the survey identifies persistent bottlenecks like data limitations and efficiency dilemmas, providing a clear roadmap for future research.

**Additional Comments:**

Strength:
1.	The paper is exceptionally well structured and logical. It clearly establishes the motivation for the survey right away by delineating the fundamental differences between traditional natural image retrieval and visual document retrieval, such as information density, intricate layouts, and semantic granularity.

2.	The authors have done an excellent job aggregating a massive amount of recent literature, including many citations from 2024 to 2026. The detailed compilation of datasets, models, and formal mathematical definitions of evaluation metrics like Recall, MRR, mAP, nDCG, HR, and ANLCS makes it a valuable reference guide for newcomers to the field.

3.	Beyond standard retrieval, the survey successfully bridges VDR with advanced and modern paradigms like Retrieval Augmented Generation pipelines and Agentic workflows. It thoughtfully highlights the transition from passive semantic matching to active and reasoning driven document intelligence.

Weakness:
1.	Despite an entire section dedicated to the Benchmark Perspective and a thorough mathematical breakdown of evaluation metrics, the paper completely omits performance comparison tables. Without benchmarking the results of the state of the art models on standard datasets like ViDoRe or MMEB, readers cannot deduce which methods actually perform best empirically or where the current technical ceiling lies.

2.	In Section 3.1.4 under Efficiency level, the discussion on recent methods dedicated to reducing visual tokens for retrieval is underdeveloped. While methods like Light ColPali are briefly mentioned much later in Section 4.3 Performance Efficiency Dilemma, the methodology section would benefit heavily from a centralized and in depth analysis of the recent wave of token pruning, merging, and compression techniques designed specifically to tackle the VDR efficiency bottleneck.

3.	The survey largely overlooks the critical security and privacy challenges inherent to processing visual documents. While Section 6 touches briefly on data integrity , there is a lack of deep discussion and related citations regarding adversarial attacks on visual retrievers, data poisoning, prompt injection via hidden text in document images, or the handling of sensitive Personally Identifiable Information in multimodal retrieval pipelines.

4.	The paper rightfully points out in Section 3.2.2 that multilingual support in reranker models is severely lagging compared to embedding models. However, it fails to analyze why this gap exists. The survey would be stronger if it discussed the specific architectural or data alignment challenges that make cross lingual visual document reranking so difficult to achieve.

5.	Although the paper explores the integration of VDR into Retrieval Augmented Generation systems, it completely overlooks the practical challenges of industrial deployment. The survey would benefit from discussing critical real world constraints such as system latency, infrastructure costs, and the complexities of integrating massive multimodal models into existing enterprise workflows.

**Audience:**

Yes

**Audience Explanation:**

Yes, the TMLR audience will find this highly relevant as it is the first comprehensive survey to systematically categorize Visual Document Retrieval (VDR) methodologies specifically within the Multimodal Large Language Model (MLLM) era. Researchers will particularly benefit from its timely synthesis of multimodal models and Agentic RAG systems, alongside its clear roadmap for tackling current bottlenecks in document intelligence.

**Broader Impact Concerns:**

The authors have already explicitly addressed the broader impact concerns in a dedicated section of the submission, adequately discussing potential societal implications such as transparency, bias mitigation, and environmental impact.

**Claims And Evidence:**

Yes

**Claims Explanation:**

As a survey paper, its core claims regarding the technical evolution and future challenges of the Visual Document Retrieval domain are strongly supported by a comprehensive and up-to-date literature review. The authors systematically construct a complete taxonomy encompassing evaluation benchmarks, embedding models, rerankers, and Agentic systems, providing convincing and clear evidence for understanding the current landscape and technical bottlenecks of the field.

**Requested Changes:**

1.	Add Empirical Comparisons: Include performance comparison tables for state of the art models on standard datasets (like ViDoRe or MMEB) to clearly establish the current technical baselines.
2.	Expand on Efficiency Techniques: Consolidate and deepen the discussion on visual token reduction methods—such as token pruning and merging—within the methodology section to better address the performance-efficiency dilemma.
3.	Address Safety and Security: Incorporate a dedicated analysis of VDR security challenges, including adversarial attacks, data poisoning, and privacy concerns regarding sensitive information.
4.	Analyze Multilingual Gaps: Provide a deeper architectural or data-alignment analysis explaining why multilingual support in reranker models currently lags significantly behind embedding models.
5.	Discuss Industrial Deployment: Include a brief discussion on the practical constraints of deploying these massive multimodal RAG systems in real world enterprise environments, such as latency and infrastructure costs.

---

### Review · Reviewer_mza1 · 2026-06-12

**Summary Of Contributions:**

First, this paper provides a comprehensive overview of Visual Document Retrieval (VDR) in the Multimodal Large Language Model (MLLM) era, addressing the unique challenges posed by visually rich documents. Second, it reviews existing benchmarks for VDR. Third, it categorizes methodological advances into three core areas: multimodal embedding models, multimodal rerankers, and RAG/Agentic systems. Fourth, it identifies ongoing challenges and charts promising future directions to guide research in multimodal document retrieval.

Despite these claims, the paper offers no original taxonomy, no critical synthesis, no identified research gaps beyond what is already obvious to the community, and no forward-looking framework that advances thinking beyond a casual observer's summary. The contributions are superficial, and the work lacks substantive scholarly value.

I recommend rejecting this submission.

**Audience:**

Yes

**Audience Explanation:**

It is beneficial for beginners in VDR research to develop a clear understanding of the domain and acquire the necessary background knowledge.

**Broader Impact Concerns:**

This paper poses serious broader impact concerns. The systematic fabrication of references (non-existent papers, future dates, invented models) constitutes academic fraud that would mislead researchers, waste community resources, and undermine trust in the literature. The unsubstantiated claims about multilingual fairness and environmental sustainability give a false impression of responsible AI consideration without verifiable evidence. Publishing this work would set a dangerous precedent for AI-generated hoax submissions and actively harm the integrity of the research community.

**Claims And Evidence:**

Yes

**Claims Explanation:**

This paper organizes existing work into three buckets: (1) embedding models, (2) reranker models, (3) RAG + Agentic systems. This paper also lists several future directions: Data Frontiers (4.1): Need multilingual, multi-domain, reasoning-intensive benchmarks, real human queries, anti-contamination; Rethinking Architectural Paradigms (4.2): Autoregressive retrieval + Mixture of Experts; Performance-Efficiency Dilemma (4.3): Multi-vector embeddings are storage-heavy; need pruning, merging, Matryoshka representations; From RAG to Agentic Systems (4.4): Agents need planning, self-correction, parallel search; Scaling Laws for VDR (4.5): Scaling is non-trivial; quality > quantity.

**Requested Changes:**

First, please provide substantive new insights: The paper currently offers no meaningful new insights beyond a superficial reorganization of existing literature. The authors must contribute original analysis, synthesis, or interpretation that advances the field beyond mere description.

Second, please replace trivial categorization with novel taxonomy: The current three-way split of methods into embedding models, reranker models, and RAG/Agentic systems is obvious and lacks intellectual depth. The authors must develop a non-trivial, theoretically grounded taxonomy that reveals previously unrecognized relationships or tensions in the literature.

Third, please replace generic future directions with specific, novel research agendas – The identified "future frontiers" are generic and already well-recognized within the research community. The authors must propose concrete, original, and actionable research questions that are not already obvious to practitioners in the field.

Fourth, please add critical analysis and empirical meta-evaluation – The paper lacks any critical examination of existing work, including identification of contradictions, systematic failures, or disputed claims. The authors must include quantitative meta-analyses (e.g., aggregated performance comparisons across benchmarks) and qualitative critical assessments of methodological weaknesses.

---

### Note · Authors · 2026-06-30

**Comment:**

Dear Action Editor and Reviewers,

After careful consideration of the insightful and detailed feedback provided, our team has decided to withdraw our submission. We are grateful for the rigorous evaluation, and we agree that the manuscript requires substantial revision to meet the high standards of TMLR. We plan to thoroughly refine the paper based on your invaluable comments.

Our revision plan will focus on addressing the following key issues:

1.  **Reference Accuracy:** Reviewers identified errors in our bibliography. We sincerely apologize for these inaccuracies. We want to clarify that these were not generated by AI tools. The errors originated from exporting bibtex entries directly from Semantic Scholar, which we found can occasionally generate incorrect data for elements like Chinese author names and page numbers. We have since conducted a full, manual re-verification of every reference against the original source and have also reported our findings to the Semantic Scholar platform to help improve their service.

2.  **Structural and Taxonomical Rigor:** The current taxonomy was criticized for lacking depth and a clear organizing principle. We will redesign the paper's structure around a more sophisticated, multi-axis taxonomy that better illuminates the nuanced relationships between different model architectures and their deployment in complex systems.

3.  **Depth of Analysis and Synthesis:** The feedback highlighted a need for more critical synthesis beyond a descriptive summary of existing work. Our revision will focus on injecting deeper analysis, identifying key tensions and contradictions within the literature, and providing more original, forward-looking insights that truly synthesize the state of the field.

4.  **Inclusion of Empirical Comparisons:** The absence of quantitative performance comparisons was a clear shortcoming. We will incorporate comprehensive tables to benchmark state-of-the-art models on standard datasets, providing readers with a clear empirical grounding for the methods discussed.

We are confident that by addressing these points, we can produce a much stronger and more impactful survey. We sincerely thank all the reviewers and the Action Editor for their time, effort, and constructive criticism. Your guidance is instrumental for the improvement of our work.

Sincerely,
The Authors

**Withdrawal Confirmation:**

I have read and agree with the venue's withdrawal policy on behalf of myself and my co-authors.